# Collateral sensitivity to pleuromutilins in vancomycin-resistant *Enterococcus faecium*

Qian Li [1,10], Shang Chen [1,10], Kui Zhu [1,2✉], Xiaoluo Huang[3], Yucheng Huang[1], Zhangqi Shen [1], Shuangyang Ding[2], Danxia Gu[4], Qiwen Yang[5], Hongli Sun[5], Fupin Hu[6], Hui Wang[7], Jiachang Cai[8], Bing Ma[9], Rong Zhang [8✉] & Jianzhong Shen [1,2✉]

The acquisition of resistance to one antibiotic sometimes leads to collateral sensitivity to a second antibiotic. Here, we show that vancomycin resistance in *Enterococcus faecium* is associated with a remarkable increase in susceptibility to pleuromutilin antibiotics (such as lefamulin), which target the bacterial ribosome. The trade-off between vancomycin and pleuromutilins is mediated by epistasis between the *van* gene cluster and *msrC*, encoding an ABC-F protein that protects bacterial ribosomes from antibiotic targeting. In mouse models of vancomycin-resistant *E. faecium* colonization and septicemia, pleuromutilin treatment reduces colonization and improves survival more effectively than standard therapy (linezolid). Our findings suggest that pleuromutilins may be useful for the treatment of vancomycin-resistant *E. faecium* infections.

[1] National Center for Veterinary Drug Safety Evaluation, College of Veterinary Medicine, China Agricultural University, Beijing 100193, China. [2] Beijing Key Laboratory of Detection Technology for Animal-Derived Food Safety, Laboratory of Quality & Safety Risk Assessment for Animal Products on Chemical Hazards (Beijing), Ministry of Agriculture and Rural Affairs, Beijing 100193, China. [3] Shenzhen Institute of Synthetic Biology, Shenzhen Institutes of Advanced Technology, Chinese Academy of Sciences, Shenzhen 518055 Guangdong, China. [4] Centre of Laboratory Medicine, Zhejiang Provincial People's Hospital, People's Hospital of Hangzhou Medical College, Hangzhou 310014, China. [5] Department of Clinical Laboratory, State Key Laboratory of Complex Severe and Rare Diseases, Peking Union Medical College Hospital, Chinese Academy of Medical Sciences and Peking Union Medical College, Beijing 100730, China. [6] Institute of Antibiotics, Huashan Hospital, Fudan University, Shanghai 200040, China. [7] Department of Clinical Laboratory, Peking University People's Hospital, Beijing 100044, China. [8] Department of Clinical Laboratory, Second Affiliated Hospital of Zhejiang University, School of Medicine, Zhejiang, Hangzhou 310009, China. [9] Clinical Laboratory, Medicine Department, Henan Provincial People's Hospital, Zhengzhou 450003, China. [10]These authors contributed equally: Qian Li, Shang Chen. ✉email: zhuk@cau.edu.cn; zhang-rong@zju.edu.cn; sjz@cau.edu.cn

The patterns of evolutionary cross-resistance to clinical antibiotics are a major driving force for accelerating the emergence and dissemination of multidrug resistant (MDR) bacteria[1]. Potential avenues for antibiotic resistance are co-selection for resistance driven by antimicrobial-producing organisms and harmful substances in natural environments where active compounds compete for the same targets[2]. The possibility of such cross-resistance to clinical antibiotics has received intensive attention previously. Antibiotic resistance often carries various fitness costs in the absence of selective pressures, and such trade-offs in turn occasionally result in rugged fitness landscapes to channel the evolutionary trajectory[3]. However, the understanding that whether and how negative responses to antibiotic selective pressure modulate the trajectory of evolution in bacteria remains unclear[4,5]. Indeed, pioneering works have recently implicated the patterns of collateral sensitivity in *Escherichia coli*[6], *Pseudomonas aeruginosa*[7], *Staphylococcus aureus*[8,9] and other pathogens, wherein resistance to one antibiotic simultaneously induces susceptibility to another by forming either homogeneous or heterogeneous populations[10] (Fig. 1a). The clinical implications of collateral sensitivity therefore supply prioritized rational therapies such as the sequential or concurrent deployment of reciprocal collateral sensitivity antibiotic pairs, to combat the antibiotic resistance crisis.

The increasing vancomycin-resistant *enterococci* (VRE) particularly vancomycin-resistant *Enterococcus faecium* (VRE_fm), with high genomic plasticity and metabolic flexibility, seriously compromise the effectiveness of existing antibiotics[11]. The persistence and spread of VRE_fm within health care settings has become one of the most challenging nosocomial pathogens, leading to at least 5400 estimated deaths and more than $500 million in excess health care costs annually in the United States (U.S.)[12] and accounting for 37% nosocomial infections in Germany[13]. Although vancomycin

resistance is widely mediated by mega-plasmids containing diverse *van* gene clusters, these resistance alleles have been persistently maintained in VRE with low or without fitness cost[14]. Therefore, a better mechanistic understanding of antibiotic resistance is urgently required to combat VRE_fm associated infections. However, the evolutionary and pharmacological consequences of dedicated *van* genes remain largely unknown. In this work, we exploit the evolutionary trade-offs to identify the collateral sensitivity patterns of 102 clinical VRE_fm isolates nationwide for antibiotic repurposing. Our observations indicate that pleuromutilin antibiotics are promising candidates targeting VRE_fm in vitro and in animal models, shedding light on the evolution-directed rational design of antibiotic therapies against MDR bacterial infections.

## Results and discussion

**VRE_fm show collateral sensitivity to pleuromutilins.** To identify the trade-offs in VRE_fm, we first determined the susceptibility and constructed the network of collateral sensitivity of 10 main classes of antibiotics routinely used in the clinic against both 20 VRE_fm and 20 vancomycin sensitive *E. faecium* (VSE_fm) isolates. Markedly, compared to VSE_fm, we observed that VRE_fm exhibited specific susceptibility to lefamulin (Fig. 1b, Supplementary Tables 1, 2), an approved pleuromutilin antibiotic by the U.S. Food and Drug Administration (FDA) for community-acquired bacterial pneumonia (CABP) in 2019[15], although they displayed general cross-resistance to multiple antibiotics as expected (Fig. 1c). The susceptibility to lefamulin increased more than 500 folds, from ≥16 μg/mL to 0.03 μg/mL. To extend whether other ribosome-targeting antibiotics display similar behaviors, we tested eight antibiotics with different binding sites[16]. It confirmed a strong collateral sensitivity to lefamulin with the minimum

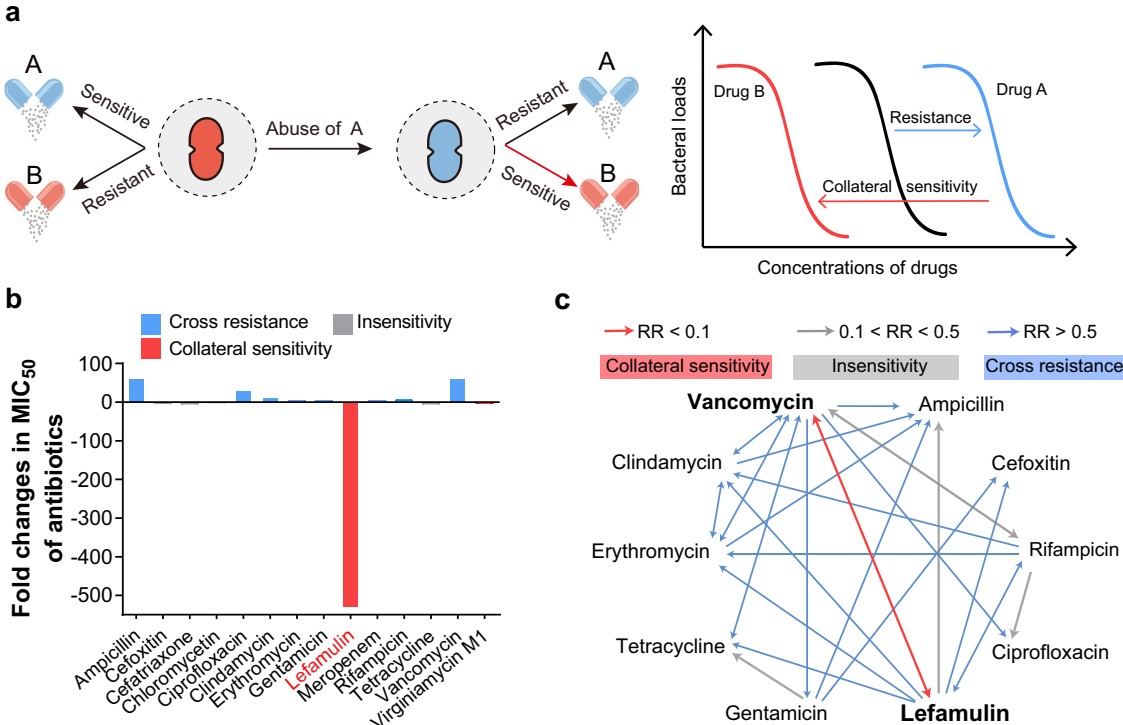

**Fig. 1 Vancomycin-resistant *E. faecium* show collateral sensitivity to lefamulin. a** Scheme of collateral sensitivity. Bacterial evolution of resistance to one antibiotic is usually accompanied by collateral sensitivity to another antibiotic. **b** Fold changes in MIC_50 of 14 antibiotics between VRE_fm (*n* = 20) and VSE_fm (*n* = 20). **c** Collateral sensitivity network among antibiotics in *E. faeicums*. For collateral sensitivity network, the directed paths of each arrow represent the collateral sensitivity (red), cross resistance (blue) and insensitivity (gray). Resistance ratio (RR) = MIC_50 (resistance)/MIC_50 (parental). Collateral sensitivity (CS): RR < 0.1, insensitivity: 0.1 < RR < 0.5, cross resistance (CR): RR > 0.5.

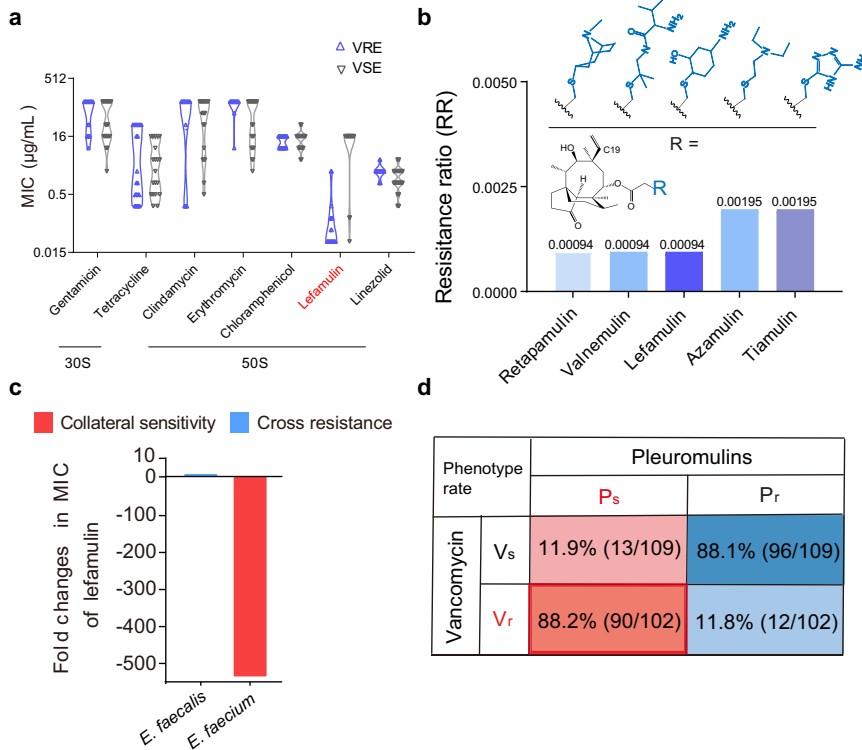

**Fig. 2 Collateral response to pleuromutilins is universal in VRE$_{fm}$. a** MIC$_{50}$ of eight ribosome-targeting antibiotics against *E. faeciums* ($n = 40$). **b** Resistance ratios of pleuromutilins against 210 *E. faecium* isolates. Structures of five pleuromutilins, where the C19 of the tricyclic mutilin core is single bond in azamulin. (**c**) Fold changes in MIC$_{50}$ of lefamulin against *E. faeciums* ($n = 210$) and *E. faecalis* ($n = 10$). **d** Proportion of four phenotypes in *E. faeciums*. ~90% VRE$_{fm}$ ($n = 102$) are sensitive to pleuromutilins, whereas ~90% of VSE$_{fm}$ ($n = 109$) are resistant. Vs: sensitivity to vancomycin, Vr: resistance to vancomycin, Ps: sensitivity to pleuromutilins, Pr: resistance to pleuromutilins.

inhibitory concentration required to inhibit the growth of 50% bacteria (MIC$_{50}$) of 0.03 µg/mL in VRE$_{fm}$, whereas most VSE$_{fm}$ were resistant to lefamulin with MIC$_{50}$ of 16 µg/mL (Fig. 2a). Therefore, we hypothesized that such collateral response could be due to the distinct modes of action of pleuromutilins (PLEs).

To test the generality of VRE$_{fm}$ channeled evolution toward elevated susceptibility to pleuromutilins, we first performed the structure-activity relationship (SAR) analysis of five pleuromutilins routinely used in human and veterinary medicine (Fig. 2b, Supplementary Fig. 2a). We expanded the number of *E. faecium* isolates to 210, including 102 VRE$_{fm}$ isolates of human origins and 109 VSE$_{fm}$ isolates of human, animals and probiotics in China. These VRE$_{fm}$ isolates validated the robust collateral sensitivity to pleuromutilins, particularly the decreased MICs of lefamulin, retapamulin and valnemulin with resistance ratios of <0.002 (Fig. 2b, Supplementary Table 1). Our results are consistent with previous empirical observations that the analogs of pleuromutilin tend to be efficacious against VRE$_{fm}$[17,18]. To further verify this phenomenon, we compared the genomes of 79 global VRE$_{fm}$ isolates (Supplementary Table 3) with 21 domestic ones, and found that the allelic profiles in China showed high genetic relationships with that in Europe and North America (Supplementary Fig. 1a). Notably, the *vanA* gene clusters are dominant (95%, 20/21) in 21 clinic VRE$_{fm}$ isolates (Supplementary Fig. 1b), consisting with the *vanA* genotype most frequently recorded[11,13]. To further assess the generality of collateral sensitivity in other *van* gene clusters, we found that a clinical *vanB*-type *E. faecium* isolate showed similar collateral patterns to five pleuromutilins (Supplementary Table 2). Then, we tested the activity of lefamulin against *E. faecalis* including vancomycin resistant/ sensitive isolates. In contrast to VRE$_{fm}$, vancomycin-resistant *E. faecalis* (VRE$_{fs}$) were resistant to lefamulin (Fig. 2c), implying the species specificity of

collateral sensitivity in *enterococci*. The collateral pattern indicates the species-specific effect[19], where the collateral sensitivity to pleuromutilins in VRE$_{fm}$ is contingent upon the intrinsic genomics. It is consistent with previous studies[20,21] that the *van* gene clusters are located on pMG1-like and pheromone sensing plasmids in VRE$_{fm}$ and VRE$_{fs}$, respectively. Additionally, the ubiquity of plasmid-encoded toxin-antitoxin gene systems may also account for the species-specific effect[22]. Collectively, it suggests that the antibiotic pairs of vancomycin and pleuromutilins exhibit ubiquitous collateral sensitivity in VRE$_{fm}$.

To further characterize the collateral response, we noticed the susceptibility of 210 *E. faeciums* isolates in four patterns that approx. 90% isolates (89/102) of VRE$_{fm}$ display collateral sensitivity to pleuromutilins using lefamulin as a model (P$_s$V$_r$) (Fig. 2d). Intriguingly, we observed the contrary pattern of pleuromutilin susceptibility in VSE$_{fm}$, in which approx. 90% isolates (96/109) of VSE$_{fm}$ were resistant to pleuromutilins (P$_r$V$_s$). Therefore, *E. faecium* shape the divergent evolution upon pleuromutilins to four patterns, including P$_s$V$_r$, P$_s$V$_s$, P$_r$V$_s$ and P$_r$V$_r$, and in turn such contingency can be used to design rational approaches to treating the prevalent VRE$_{fm}$ associated infections.

**Heterogeneous collateral responses are channeled by ribosomes.** The unique collateral sensitive antibiotics suggest that the interaction between pleuromutilins and the target may shed light on elucidating the evolutionary conservation in VSE$_{fm}$. First, lefamulin showed moderate bacteriostatic activity against VRE$_{fm}$ at a high level (1.2 µg/mL, 40 × MIC) (Supplementary Fig. 3). Furthermore, we quantified the accumulation of lefamulin in eight *E. faecium* isolates including all four phenotypes of heterogeneous collateral responses and found no difference (Supplementary Fig. 5a, b). Last, we confirmed there was almost no

further modifications for the accumulated lefamulin in *E. faecium* (Supplementary Fig. 5c–j). These data denote that the intrinsic properties of pleuromutilins should be excluded and the main target of pleuromutilins should eventually dominate the trajectory of collateral response in VRE_fm.

Pleuromutilins exhibit distinctive recognition to the peptidyl transferase center (PTC) by blocking bacterial protein biosynthesis[16]. Pleuromutilins mainly interact with eight nucleotides in the PTC domain. We first confirmed that there were no mutations at such sites in 20 VRE_fm and 20 VSE_fm isolates based on whole-genome sequence analysis (Fig. 4a, Supplementary Figs. 2c, 6b). Moreover, we found no genes such as *cfr*[23] encoding methyltransferases to confer universal resistance (Supplementary Fig. 7a–c). Interestingly, we found that pleuromutilins had a higher affinity to ribosomes in $P_sV_r$ than $P_rV_s$ type strains through fluorescence polarization analysis[24] (Supplementary Fig. 8a, b), implying that VRE_fm may modulate the susceptibility to pleuromutilins through the mechanism of ribosomal protection. The ATP-binding cassette F (ABC-F) protein family protects bacterial ribosomes from multiple classes of ribosome-targeting antibiotics[25]. We found the presence of four prevailing genes (*msrC*, *eatA/eatAv* and *lsaE*) of the 26 members of the ABC-F family[26] in 40 *E. faecium* isolates (Supplementary Fig. 6b). Nevertheless, both *lsaE* and *eatA/eatAv* were partially carried by clinical isolates and expressed at similar levels in the presence and absence of pleuromutilins (Supplementary Fig. 9a, b). Extremely, we noticed that the species-specific gene *msrC*[27] expressed more than 70-fold higher in model strain VSE_fm CAU310 than VRE_fm CAU369 (Supplementary Fig. 9c), suggesting that the low expression of *msrC* may potentiate the efficacy of pleuromutilins. Meanwhile, we observed that the decreased transcription of *msrC* in two lefamulin sensitive VRE_fm isolates (Fig. 3a, Supplementary Fig. 10a) is in a dose-dependent manner of lefamulin. Furthermore, we found the increased transcription of *msrC* in all 12 lefamulin resistant isolates as well (Fig. 3b, Supplementary Figs. 10b, 11). Constantly, we obtained the increased expression of MsrC in a lefamulin resistant VRE_fm CAU378 treated with lefamulin for 1 h (Fig. 3c), based on proteomics analysis. Additionally, the MsrC overexpression strain was constructed in a pleuromutilins-sensitive strain using conjugative transformation (Supplementary Fig. 12a). We found that the conjugant (pAM401 + *msrC*) shows high expression of *msrC* and are resistant to all pleuromutilins (Supplementary Fig. 12b–d). Taken together, these results indicate that *msrC* is linked with the collateral sensitivity in VRE_fm.

To explore how MsrC reduces the susceptibility to pleuromutilins, we performed a simulation analysis on the interaction between MsrC and the PTC domain[28]. Compared to the affinity between pleuromutilins and PTC with Z-docker interaction energy of −146.324 kcal, the residues of R241 and L242, and K233 and K246 in MsrC competitively bound to the shared binding sites U2504 and C2063 of MsrC and lefamulin in PTC domain, respectively, with Z-docker interaction energy of −170.611 kcal (Fig. 3d). These findings indicate that high expression of species-specific MsrC tends to block pleuromutilins targeting PTC in VSE_fm, whereas the low expression of MsrC facilitate the binding of pleuromutilins in VRE_fm accordingly.

**Epistasis between *van* gene clusters and *msrC*.** To understand how *msrC* mediates collateral response in VRE_fm, we first observed the delayed growth curves of a constructed conjugant containing the *vanA*-plasmid from *E. faecium* CAU369 (pCAU369) in *E. faecium* BM4105-RF[20], in the presence of sub-inhibitory levels of lefamulin (Supplementary Fig. 14b, c). It suggested that the plasmid carrying the *van* gene cluster plays a crucial role in the increased susceptibility

to pleuromutilins in VRE_fm. Subsequently, we showed that most *van* gene clusters are dominant *vanA*-type (95%, 19/20) (Supplementary Fig. 6b), consisting with previous reports that the increasing dissemination of *vanA*-VRE_fm is prevalent world widely, particularly in the U.S. and Europe[29,30]. Therefore, we compared 14 plasmids containing *vanA* gene clusters in global *E. faecium* isolates (Supplementary Table 5) with that in *E. faecium* CAU369, and noticed only the *vanA* gene clusters were present in all isolates (Supplementary Fig. 13). Thus, we deduced that the *vanA* gene clusters (~ 6–7 kbp), instead of the other motifs in the *vanA*-type megaplasmids[31] (~ 30–150 kbp), contribute to the collateral sensitivity to pleuromutilins in VRE_fm.

Given that the decreased expression of *msrC* in VRE_fm (Fig. 3a, Supplementary Fig. 9c), we hypothesized a negative feedback between *msrC* and the *vanA* gene clusters. To verify such epistasis, we found the sequence in the promoter of *msrC* shared high similarity (83.3%, 10/12) to the promoters of *vanR/H/Y*[32] (Fig. 4a) and the promoters are conserved in both *vanA*- and *vanB*-type isolates globally (Supplementary Fig. 15), indicating that phosphorylated VanR may simultaneously induce *van* transcription but inhibit *msrC* transcription. Toward this goal, we exploited transcriptome analysis on VRE_fm treated with lefamulin. Remarkably, we observed *vanRS* activation and decreased *msrC* transcription in pleuromutilin-sensitive VRE_fm CAU369, whereas there was no *vanS* activation and 33.1% *vanR* transcription, and increased *msrC* transcription in pleuromutilin-resistant VRE_fm CAU378 (Supplementary Fig. 16). Consistently, we confirmed the opposite patterns of *vanS* and *msrC* transcription in pleuromutilin-resistant isolates under lefamulin treatments based on qRT-PCR and proteomics analysis (Fig. 4b–d, Supplementary Fig. 17a, b). To validate that *vanR* modulates *msrC* expression, we constructed a conjugant by receiving a recombinant *vanRS* plasmid in a pleuromutilin resistant *E. faecium* (Supplementary Fig. 18). The transcription of *vanR* and *vanS* were activated in a dose-dependent manner under lefamulin treatments (Supplementary Fig. 19a, b), in turn, the transcription of *msrC* in the conjugant (pAM401 + *vanRS*) was dramatically inhibited (Supplementary Fig. 19c). Correspondingly, the conjugant with *vanRS* expression is sensitive to pleuromutilins, with more than 16-fold decreased MICs (Supplementary Fig. 19d). These results support our hypothesis that the negative epistasis between the *van* gene cluster and *msrC* is responsible for the collateral sensitivity to pleuromutilins in VRE_fm.

To further verify that P-VanR regulates *msrC* transcription, we first demonstrated that P-VanR binds to the *msrC* promoter using electrophoretic mobility shift assay (Fig. 4e, f). Furthermore, we calculated the EC_50 (effective concentration for 50% response) as 1.01 μmol/L for P-VanR binding to the fragment of *msrC* promoter, which is much lower than the binding affinity between P-VanR and the *vanH* promoter (Fig. 4g). When in the presence of inducers such as classic vancomycin and lefamulin, VanS switches its activity from phosphatase to kinase, phosphorylating the cognate response regulator VanR[33]. Phospho-VanR then binds to similar promoter sequences triggering the transcription of the *van* gene cluster, and inhibits *msrC* transcription to induce VRE_fm sensitive to pleuromutilins accordingly (Fig. 4h). Altogether, these results indicate that P-VanR co-regulates *vanR/vanH* and *msrC* by binding to similar promoter fragments, to facilitate pleuromutilins against VRE_fm.

In addition, antibiotic resistance/tolerance can be regulated by metabolic reprogramming[34,35], which, in principle, could boost the effectiveness of antibiotics, consistent with the observation that anaerobic glycolysis further exacerbated the growth of VRE_fm with *vanA*-plasmid in the presence of subinhibitory levels of lefamulin (Supplementary Fig. 14c). To test this possibility, we dissect whether metabolites modulate collateral response in VRE_fm. Since there is no Krebs cycle in *enterococci*[36], the

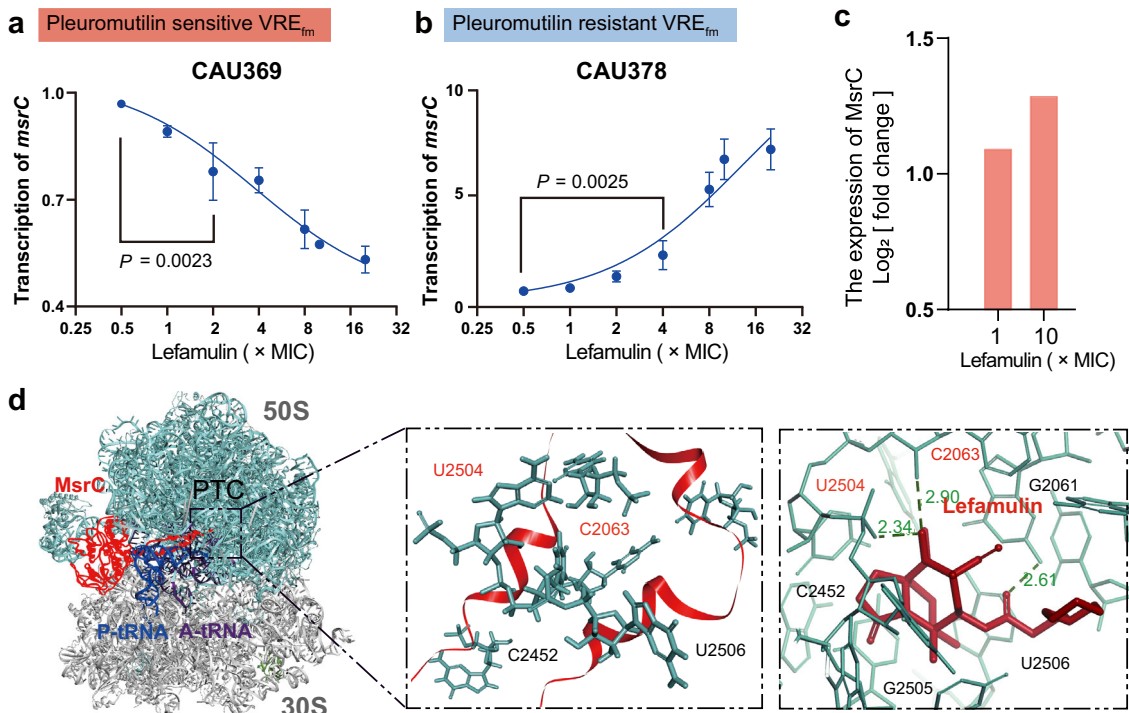

**Fig. 3 Low expression of *msrC* is linked with collateral sensitivity. a–b** Transcription analysis of *msrC* in pleuromutilin-sensitive VRE_fm CAU369 (**a**) and pleuromutilin-resistant VRE_fm CAU378 (**b**). **c** Proteomics analysis of VRE_fm CAU378 treated with 1× and 10 × MIC lefamulin for 1 h. Proteins were identified as significantly different with fold changes of log_2 [fold changes] values of at fold-increase or fold-decrease of expression levels. **d** Comparison of lefamulin binding sites within the MsrC-ribosome complex. The 50S (green) and 30S (gray) subunits (PDB 5AA0 [https://www.rcsb.org/structure/5AA0]), and the A-site tRNA (purple) and P-site tRNA (blue) are shown. Binding of MsrC (red, homology of the template with PDB 5ZLU [https://www.rcsb.org/structure/5ZLU], middle) blocks the binding of lefamulin (brick-red, PDB 5HL7 [https://www.rcsb.org/structure/5HL7], right) to PTC in the 50S subunit. Data were presented as means ± S.D. n.s., not significant, determined by non-parametric one-way ANOVA (*n* = 3).

dominant pyruvate metabolism kept steadily (Supplementary Fig. 21a–c). Considering the abundance of metabolic versatility and the challenge to genetically manipulate clinical isolates, its contribution to collateral sensitivity in most VRE_fm with mega-plasmids remains unclear.

**Collateral sensitivity in vivo.** Given that VRE usually resists multiple classes of antibiotics, we hypothesized that pleuromutilins might be used to treat VRE associated infections based on our observation. Lefamulin was recently approved in the U.S. and European Union for the treatment of CABP in adults. To further assess the efficacy of lefamulin, we empirically evaluated its efficacy in two mouse models including an intestinal coloni-zation model (VRE_fm CAU369) and in a peritonitis-septicemia model in mice (VRE_fm CAU427) (Fig. 5a). First, we found that VRE_fm CAU369 mainly colonized in the cecum and colon with remarkably decreased bacterial burden after a single dose of lefamulin (Fig. 5b). Remarkably, lefamulin showed better anti-bacterial efficacy than linezolid, the only antibiotic with the U.S. Food and Drug Administration (FDA) approval for treating VRE infections, especially in the cecum and colon on the first day after administrations. Meanwhile, we collected daily the feces for bacterial counting. Compared to clinically recommended line-zolid, lefamulin similarly produced time-dependent reduction in bacterial burden of feces over the 7-day treatment period (Fig. 5c). In addition, VRE_fm promptly dominated mouse gut microbiota after intragastric inoculum based on bacterial com-munity analysis, whereas lefamulin administration notably reduced its abundance through the upregulated the genera of *Akkermansia* and *Klebsiella* (Fig. 5d). After 7 days, all mice nearly eliminated the pathogens and appeared healthy. Intriguingly,

lefamulin could restore the homeostasis of fecal microbiota faster than the untreated group according to the Chao1 and Shannon indexes (Supplementary Fig. 22a, b), particularly the patterns of *Bacteroides*, *Prevotella*, *Clostridium*, and *Escherichia/Shigella* on the 7th day (Supplementary Fig. 22c–f). In the peritonitis-septicemia model, the survival curve showed that all mice sur-vived under the treatment of a dose of 10 mg/kg lefamulin after 96 h (Fig. 5e), with reduced bacterial counts in organs (Fig. 5f). Altogether, our results suggest that lefamulin efficiently remedies the severity of VRE_fm infections, which may be extended to treat and diminish bacterial colonization since that there are limited effective agents available to treat such infections in clinic.

For the foreseeable future, VRE especially VRE_fm will remain an important nosocomial problem[37,38], particularly the steady increase in the U.S. since the early 1990s in hospitals[11]. Because VRE_fm has developed resistance to essentially every antibiotic used in clinic, novel therapeutic strategies to explore collateral sensitivity may be attempted. A better understanding of the evolutionary biology particularly epistasis by which the *van* genes manipulate ribosome protection could shed light on intervention strategies against VRE_fm pathogens. Notably, plasmid mediated epistasis is not restricted to *E. faecium*, and we postulate that such collateral response may be widespread phenomenon in other bacteria. Mobile plasmids drive the spread of many critical antibiotic resistance genes in clinical pathogens[39]. Most recently, it has been demonstrated the collateral sensitivity associated with transferable plasmids in clinical *E. coli* isolates[40], compared to previous empirical observations and mutations in chromosome and plasmids[5–9]. Our results suggest that the evolutionarily collateral response has already dominated the prevalent VRE_fm carrying conjugative plasmids (Supplementary Fig. 14a). To argue

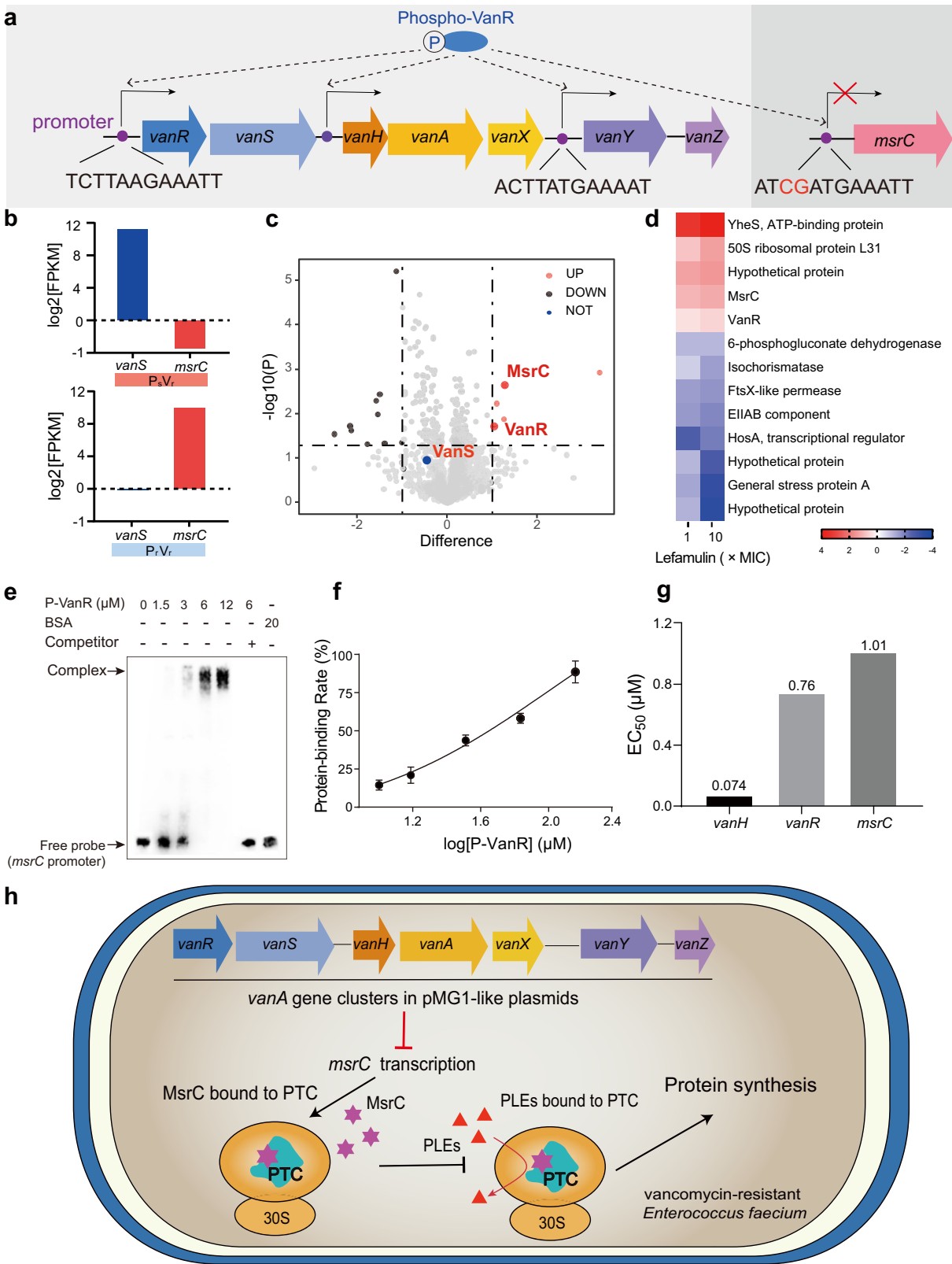

for preclinical development of pleuromutilins as leads against VRE_fm, continued studies should further focus on the evaluation whether pleuromutilins reduce VRE_fm colonization in patients and maintain microbiota recovery after its expansion.

In conclusion, our results demonstrate the epistasis between *van* genes and *msrC* mediating collateral sensitivity in most clinically relevant VRE_fm strains, to potentiate the efficacy of pleuromutilins. These observations in vitro and in vivo provide proof-of-concept for an efficient therapeutic option against the increasingly aggravating VRE_fm associated infections by revitalizing existing antibiotics. Pleuromutilins not only exhibit robust bacteriostatic activity against clinical isolates nationwide but also attenuate the colonization possibly improving clinical outcomes. Overall, our work expands the knowledge of evolutionary

**Fig. 4 Epistasis between the *van* gene cluster and *msrC*. a** Promoters in genes *vanR*, *vanH*, *vanY* and *msrC*. **b** Fold changes in log$_2$ [FPKM] values of relative expression of *vanS* and *msrC*. Transcriptome analysis of *E. faeciums* under the treatment of lefamulin at levels of 1× and 10 × MIC for 1 h. **c** Volcano plot represents the protein expression ratios of lefamulin treated bacterial cells (VRE$_{fm}$ CAU378). For each protein, the -log 10 (P-value) is plotted against its log$_2$ (fold change). Proteins upregulated (P < 0.05, fold change > 2) in 1× and 10 × MIC lefamulin treated samples are colored in red, proteins downregulated (P < 0.05, fold change < −2) are colored in blue, while unchanged in black. **d** Proteomics analysis of VRE$_{fm}$ CAU378 treated with 1× and 10 × MIC lefamulin for 1 h. Proteins were identified as significantly different with fold changes of log$_2$ [fold changes] values of at fold-increase or fold-decrease of expression levels. **e** Binding reactions between P-VanR (12–1.5 μM) and *msrC* promoter fragment (212-bp, 0.3 ng) based on the gel electrophoretic mobility shift assay. Free probe: biotin-labeled promoter; Competitor: Unlabeled promoter; BSA: Bovine serum albumin. Experiments were performed as three biologically independent experiments. **f** Calculated protein-binding rates of P-VanR and *vanH/vanR/msrC* promoters, based on the gray values in Fig. 4e. **g** Binding constants of *vanH/vanR/msrC* promoters and P-VanR. **h** Scheme of collateral sensitivity in VRE$_{fm}$. The decreased transcription of *msrC* by *van* genes enhances ribosome-targeting pleuromutilins binding to PTC to block protein synthesis. Data were presented as means ± S.D. n.s., not significant, determined by non-parametric one-way ANOVA (n = 3).

compensatory phenomenon for the implementation of existing antibiotics to combat MDR bacterial pathogens.

## Methods

**Antimicrobial-susceptibility test**. Minimum inhibitory concentrations (MICs) of antibiotics were performed using the standard broth microdilution method, according to the CLSI 2020 guideline. Briefly, single bacterial colonies (including *E. faecalis* ATCC 29212 and 210 *E. faeciums* isolates) were cultured in Mueller Hinton Broth (MHB) at 37 °C at 220 r.p.m. for 8–12 h. Subsequently, antibiotics with two-fold dilution in MHB were mixed with an equal volume of bacterial suspensions in MHB containing approximately 1.5 × 10$^6$ colony-forming units (CFUs)/mL in a clear UV-sterilized 96-well microtiter plate. The plate was placed in the incubator for 16–18 h at 37 °C and then the MIC values were read. MIC values were defined as the lowest concentrations of antibiotics with no visible growth of bacteria.

**Time-dependent killing assay**. An overnight culture of VRE$_{fm}$ CAU369 was diluted 1:100 in MHB and incubated at 37 °C, 200 r.p.m. for 4 h, to obtain bacteria at exponential phase. Then the bacterial suspension was challenged with lefamulin at 4 × MIC, 10 × MIC and 40 × MIC. Subsequently, 10-fold serially diluted suspensions were plated on BHI agar plates for overnight incubation at 37 °C. Finally, colonies were counted and CFUs per mL were calculated accordingly.

**Membrane integrity assay**. An overnight culture of VRE$_{fm}$ CAU369 were inoculated in 0.01 M phosphate-buffered saline (PBS) (pH 7.4) to adjust bacterial suspensions to approximately an OD600 of 0.5, following by the addition of propidium iodide (PI) to a final concentration of 10 nM, then treated with lefamulin and linezolid at the levels of 1 × MIC, 5 × MIC and 10 × MIC for 34 min after centrifugation. The fluorescence was measured at the excitation wavelength of 535 nm and emission wavelength of 615 nm in two-min intervals using the Infinite M200 Microplate reader (Tecan).

**Membrane fluidity assay**. An overnight culture of VRE$_{fm}$ CAU369 was incubated with 10 μM Laurdan at 37 °C for 20 min in the dark. The stained bacterial cell was washed with PBS for three times. Then, 190 μL of the culture was mixed with 10 μL of PBS containing lefamulin at the level of 1 × MIC, 5 × MIC and 10 × MIC in the black, clear-bottom, 96-well plates. Benzyl alcohol (50 mM) was used as a positive control. Last, the fluorescence intensities of Laurdan were measured at the excitation wavelength of 350 nm and emission wavelengths of 435 nm and 490 nm in two-min intervals using the Infinite M200 Microplate reader (Tecan). The Laurdan GP was calculated using the formula GP = (I$_{435}$− I$_{490}$)/ (I$_{435}$ + I$_{490}$).

**ROS measurement**. An overnight culture VRE$_{fm}$ CAU369 grown from a single colony in Brain Heart Infusion (BHI) broth were washed and resuspended in 0.01 M of PBS (pH 7.4) to obtain an OD600 of 0.5. Subsequently, 2′,7′-dichloro-fluorescein diacetate (DCFH-DA) was added to a final concentration of 10 μM and the mixture was incubated at 37 °C for 30 min. After washing with 0.01 M of PBS for three times, 190 μl of probe-labeled bacterial cells were added to a 96-well plate treated with 10 μl of lefamulin at the level of 1 × MIC, 5 × MIC and 10 × MIC. After incubation for another 30 min, fluorescence intensity was immediately measured with the excitation wavelength at 488 nm and the emission wavelength at 525 nm in two-min intervals using the Infinite M200 Microplate reader (Tecan).

**Membrane depolarization assay**. An overnight culture of VRE$_{fm}$ CAU369 bacterial cells were washed and resuspended to obtain an OD600 of 0.5 with 5 mM of HEPES (pH 7.0, + 5 mM of glucose). Then 3,3-dipropylthiadicarbocyanine iodide DiSC3(5) (Aladdin, ≥98%, 1 μM) was added. The dissipated membrane potential of VRE$_{fm}$ CAU369 in the presence and absence of lefamulin at the level of 1 × MIC, 5 × MIC and 10 × MIC were measured at the excitation wavelength of 622 nm and emission wavelength of 670 nm using the Infinite M200 Microplate reader (Tecan).

**△pH measurement**. △pH was measured in the presence of pH-sensitive fluorescent probe BCECF-AM (10 μM). An overnight culture of VRE$_{fm}$ CAU369 bacterial cells treated with lefamulin at the level of 1 × MIC, 5 × MIC and 10 × MIC were measured at the excitation wavelength of 500 nm and emission wavelength of 522 nm for 40 min, using the Infinite M200 Microplate reader (Tecan).

**ATP determination**. Extracellular and intracellular levels of ATP were determined using an Enhanced ATP Assay Kit (Beyotime). VRE$_{fm}$ CAU369 grown overnight at 37 °C with shaking at 200 r.p.m. were washed and resuspended to obtain an OD600 of 0.5 with 0.01 M of PBS (pH 7.4). After treatment with 1 × MIC, 5 × MIC and 10 × MIC lefamulin for 1 h, bacterial cultures were centrifuged at 12,000 r.p.m. and 4 °C for 5 min, and the supernatants were collected for the determination of extracellular ATP levels. Meanwhile, bacterial precipitates were lysed by lysozyme, and centrifuged, then the supernatants were prepared for the measurement of intracellular ATP levels. The detecting solution was added to a 96-well plate and incubated at room temperature for another 5 min. Last, the supernatants were added to the well and mixed quickly, before recording in the model of luminescence using the Infinite M200 Microplate reader (Tecan).

**Antibiotic accumulation test**. The accumulation of intracellular antibiotics in bacteria was determined based on the established liquid chromatography with tandem mass spectrometry (LC-MS/MS) method[41]. Overnight cultures of bacterial cells (VRE$_{fm}$ CAU369, VRE$_{fm}$ CAU372, VRE$_{fm}$ CAU378, VRE$_{fm}$ CAU419, VSE$_{fm}$ CAU259, VSE$_{fm}$ CAU277, VRE$_{fm}$ CAU309, VRE$_{fm}$ CAU310, *S. aureus* 29213, MASA T144, *E. faecalis* 29212, VRE CAU 475) were diluted to 100 mL fresh BHI broth at 1:100 and resuspended to OD600 of 0.5 at 37 °C. Bacteria were then centrifuged at 3,000 g for 10 mins at 4 °C and the supernatants were collected for three times. Subsequently, bacterial cells were diluted to 10$^{10}$ CFU per mL with fresh PBS and aliquoted in 1.5 mL tubes. The aliquots were treated with sub-inhibitory levels of lefamulin at 37 °C for 1 h and the samples were collected to destroy the cell envelopes for LC-MS/MS analysis (Waters 2695). Additionally, structure elucidation of accumulated lefamulin in four phenotypes of *E. faeciums* isolates were based on UHPLC-Q-Orbitrap analysis (UHPLC-Q-Exactive Plus, Thermo Fisher Scientific). Extracted ion chromatogram of lefamulin obtained in the positive ESI mode at retention time of 7.08 min. Elucidation of lefamulin and its product ions based on the molecular weights of m/z 508.31 (lefamulin), and product ions of m/z 188.07 and 206.08.

**Whole-genome sequencing**. Genomic DNA were extracted from the overnight culture of *E. faecium* (including 20 VRE$_{fm}$ and 20 VSE$_{fm}$) isolates in BHI, according to the manufacturer instructions (Genome Extraction Kit, Magen). Obtained DNA was sequenced by Illumina Seq and the whole-genomes were aligned with the resistance gene database from the Center for Genomic Epidemiology (CGE).

**Molecular simulation**. Models of the complexes of lefamulin-ribosome and MsrC-ribosome were built on Discovery Studio 2018 Client. The model of MsrC was generated using the sequence from VRE$_{fm}$ CAU369. The receptor-ligand interaction and the Z-Docker reaction energy in the 50S and 30S subunits, and tRNA were performed according to a previous study[28].

**Methylation modification analysis**. Methylation modification analysis is based on the SELECT approach according to a previous study[42]. Briefly, total RNA (1500 ng) was mixed with 40 nM up primer, 40 nM down primer and 5 M dNTP in 17 μL CutSmart buffer. SELECT qPCR was performed with the following program: 90 °C for 1 min, 80 °C for 1 min, 70 °C for 1 min, 60 °C for 1 min, 50 °C for 1 min and 40 °C for 6 min. Afterward, the qRT-PCR was performed using SYBR premix Ex Taq qPCR Kit (TaKaRa). qRT-PCR was performed with the following program: 95 °C, 5 min; 95 °C, 10 s then 60 °C, 35 s for 40 cycles; 95 °C, 15 s; 60 °C, 1 min; 95 °C, 15 s. Primers for SELECT qPCR or qRT-PCR are listed in Table S6, respectively. Ct values of samples were normalized to their corresponding Ct values of control. All assays were performed with three independent experiments.

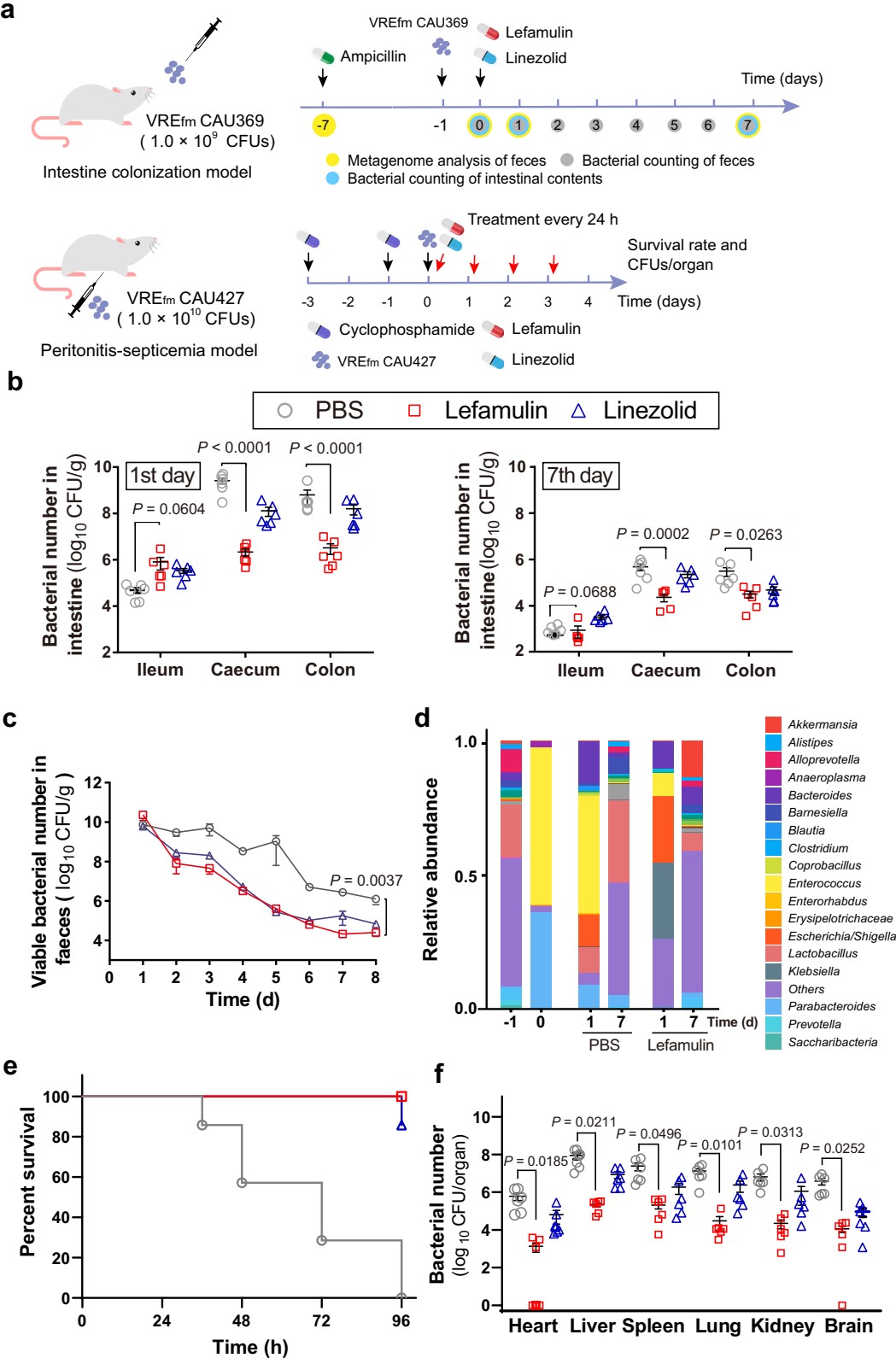

**Fluorescence polarization assay.** The fluorescence polarization value (FP value) was detected as described previously[43]. Briefly, the assay was conducted by adding 70 µL per well of VAL-DTAF (valnemulin-([4,6-dichlorotriazine-2-yl] amino)-fluorescein) tracer working solution at 0.5 µM and 70 µL per well of prepared ribosome at the level 0–2 nM to the microplate, followed by the addition of 70 µL per well of liquid A buffer (10 mM Tris [pH 7.5], 60 mM KCl, 10 mM NH₄Cl, 300 mM NaCl, 6 mM MgCl₂, 0.1 mM ATP). After the mixture was shaken for 10 s in the microplate reader, FP values were measured with the excitation wavelength

at 485 nm and the emission wavelength at 530 nm using the Infinite M200 Microplate reader (Tecan).

**Growth dynamics.** Overnight cultures of *E. faeciums* BM4105-RF and conjugant BM369-1 grown from a single colony in BHI broth were diluted 1:100 in MHB and made the final concentration approximately 1× 10⁶ CFU per mL. Lefamulin (1/8 × MIC, 1/4 × MIC, 1/2 × MIC and 1 × MIC) was added into 96 well microplates and mixed with an equal volume of cultures. Growth curves were established by OD600

**Fig. 5 Efficacy of lefamulin against VRE$_{fm}$ in vivo. a** Scheme of mouse intestinal colonization model and VRE$_{fm}$ peritonitis-septicemia model. **b** Bacterial loads in the ileum, cecum and colon. Mice ($n = 6$ per group) were given $1 \times 10^9$ CFUs of VRE$_{fm}$ CAU369, under the treatment of a single dose of 5 mg/kg lefamulin or linezolid. **c** Bacterial loads in feces. Mice ($n = 6$ per group) were given $1 \times 10^9$ CFUs of VRE$_{fm}$ CAU369 by oral gavage, with administration of 5 mg/kg lefamulin or linezolid. **d** Fecal microbiota was profiled of mice infected with VRE$_{fm}$ CAU369 in the presence of 5 mg/kg lefamulin at different points using 16S rRNA gene sequencing. **e** Survival rates of peritonitis-septicemia mice infected with VRE$_{fm}$ CAU427 ($1.0 \times 10^{10}$ CFUs) in the presence of lefamulin (10 mg/kg). **f** Lefamulin decreased bacterial loads in different organs in the mouse peritonitis-septicemia model ($n = 6$ per group). Data in **b**, **c** and **d** were presented as mean values ± S.D, **f** was presented as mean values ± SEM. *P*-values in **b**, **c** and **d** were determined by non-parametric one-way ANOVA and in **f** was determined by two-tailed t-test.

determination with an interval of 1 h at 37 °C, by Infinite M200 Microplate reader (Tecan).

**Conjugation assay**. Filter mating was performed using VRE$_{fm}$ CAU369 as the donor and *E. faecium* BM4105 (resistant to rifampicin and fusidic acid) as the recipient, according to previous study[22]. The mating mixtures of donor and recipient were filtered through a sterile membrane filter (pore size, 0.45 μm). The filters were incubated on BHI agar plates for 24 h at 37 °C. After mating, the conjugants were selected on BHI agar containing rifampin (100 μg/mL), fusidic acid (20 μg/mL) and vancomycin (20 μg/mL) after incubated at 37 °C for 24–48 h.

***msrC* overexpression mutant**. The *msrC* overexpression mutant was constructed based on a modified conjugative transformation assay, according to previous study[22]. The overexpression plasmid pAM401+*msrC* was recombined by the linearized pAM401 and the amplification of *msrC* sequence. *msrC* gene was amplified from the genome of *E. faecium* BM4105 by PCR reaction. The overexpression plasmid pAM401+*msrC* was transferred into *E. faecalis* JH2-2. Filter mating was performed using *E. faecalis* JH2-2 (pAM401 + *msrC*) as the donor and *E. faecium* GE-1 as the recipient. After mating, the conjugants were selected on BHI agar containing chloramphenicol (50 μg/mL) and arabinose after incubated at 37 °C for 24–48 h.

**Isogenic mutant constuaction**. The *vanRS* isogenic mutant was constructed based on a modified conjugative transformation assay, according to previous study[22]. The plasmid pAM401+*vanRS* was recombined by the linearized pAM401 and the amplification of *vanRS* sequence. *vanRS* gene was amplified from the genome of VRE$_{fm}$ CAU369 by PCR reaction. The plasmid pAM401+*vanRS* was transferred into *E. faecalis* JH2-2. Filter mating was performed using *E. faecalis* JH2-2 (pAM401 + *vanRS*) as the donor and *E. faecium* BM4105 as the recipient. After mating, the conjugants were selected on BHI agar containing chloramphenicol (50 μg/mL) and arabinose after incubated at 37 °C for 24–48 h.

**qRT-PCR assay**. An overnight culture of VRE$_{fm}$ CAU369 was diluted 1:100 in BHI and incubated at 37 °C, 200 r.p.m. for 4 h. Bacteria were obtained at exponential phase and treated with or without lefamulin and vancomycin (0.5 × MIC, 1 × MIC, 10 × MIC) for 1 h at 37 °C. Total RNA was extracted using the EASYspin Plus kit (Aidlab, cat. RN4301) and quantified by the ratio of absorbance (260 nm/280 nm) using a Nanodrop spectrophotometer (Thermo Scientific, MA, USA). The qRT-PCR was performed using SYBR premix Ex Taq qPCR Kit (TaKaRa, catalog no. RR820A). Cycling conditions consisted of an initial denaturation step at 95 °C for 30 s, followed by 40 cycles at 95 °C for 5 s, 60 °C for 30 s, and 72 °C for 30 s. All samples were analyzed in triplicate and the gene 16 S rRNA was used as an endogenous control as described in our previous publication[44]. The fold changes of gene expression were analyzed by $2^{-\Delta\Delta CT}$ method. Genes names and primer sequences used in the qRT-PCR analysis are listed in Table S7. GraphPad Prism 8 was used to plot graphs

**Transcriptome analysis**. Overnight cultures (VRE$_{fm}$ CAU369 and VRE$_{fm}$ CAU378) were diluted 1:100 in BHI and incubated at 37 °C, 200 r.p.m. for 4 h, to obtain bacteria at exponential phase. Then the bacterial suspensions were treated with lefamulin at the level of 1 × MIC and 10 × MIC for 15 min and 1 h. Then the bacteria were washed with fresh PBS (0.01 M, pH 7.4) three times and the bacterial cells were spun down by centrifuging at 8,000 r.p.m. for 10 min at 4 °C. RNA-Seq library building and RNA-seq data analysis are described as described previously[45].

**Metabolomics analysis**. Metabolomics were determined based on UPLC-QTOF MS analysis. Overnight cultures (VRE$_{fm}$ CAU369 and VSE$_{fm}$ CAU309) grown in BHI broth were harvested and washed with 0.01 M PBS (pH 7.4) three times. Then, approximately 50 mg precipitation of bacteria was mixed with 0.5 mL 80% methanol containing 0.1% formic acid (FA). After shaking for 30 s and ultrasonic treatment for 20 min, all samples were frozen for 1 h at −20 °C for protein precipitation. The samples were then centrifuged at 10,000 × g and 4 °C for

10 min and the supernatant (200 μL) was obtained. (O)PLS-DA (Orthogonal signal correction partial least square discrimination analysis) was performed and VIP (variable importance in projection) was calculated by MetaboAnalyst 4.0 and R project.

**Proteomics analysis**. Overnight cultures VRE$_{fm}$ CAU378 were diluted 1:100 in BHI and incubated at 37 °C, 200 r.p.m. for 4 h, to obtain bacteria at exponential phase. Then the bacterial suspension were treated with lefamulin at the level of 1 × MIC and 10 × MIC for 1 h. Then cells were lysed and the protein was reduced and alkylated by heating at 95 °C for 5 min followed by sonication for 20 min. The resulting mixture was diluted 10 times with the dilution buffer composed of 25 mM Tris–HCl (pH 8.5) and 10% acetonitrile (ACN) and the protein was digested with trypsin at 37 °C overnight. After digestion, the total protein was acidated with 5% TFA and loaded onto a C18-column which had been pretreated and equili- brated with methanol and 0.2% acetic acid respectively. For MS analysis, total peptides were dissolved in 0.1% FA and analyzed by Orbitrap Fusion Lumos mass spectrometer (Thermo Scientific). Raw data were processed with PD software package against the sequence data of VRE$_{fm}$.

**Electrophoretic mobility shift assay**. VanR purified and P-VanR prepared according to a previously described protocol[46]. Binding reactions between P-VanR (12–1.5 μM) and *msrC* promoter fragment (212-bp, 0.3 ng) based on the gel electrophoretic mobility shift assay. Briefly, P-VanR was incubated with different concentrations. After 15 min incubation, the reaction mixture was subjected to 5% nondenaturing polyacrylamide gel and electrophoresis was performed at 80 V in ice-cold bath. The images were visualized and acquired by the PharosFX imaging system (Bio-Rad, CA, USA).

**Intestinal colonization model**. To evaluate the in vivo efficacy of lefamulin, a mouse intestinal colonization model was involved according to a previous publication[44]. Briefly, 6–8 weeks old female ICR mice (20 g, $n = 6$ per group) were treated with PBS, lefamulin and linezolid. First, mice were administered 0.5 g per L ampicillin in drinking water for 5 days. On day 0, mice were infected with VRE$_{fm}$ CAU369 ($1 \times 10^9$ CFUs) by gavage. After the treatments with drugs, CFUs of VRE$_{fm}$ CAU369 were enumerated in the feces from 1 to 7 days. Meanwhile, intestinal contents (including ileum, cecum and colon) were collected on the first and seventh day for bacterial counting and the evaluation of the abundance and diversity of intestinal flora, using a previously described method[44]. The alpha analysis and the beta diversity analysis were performed based on PCoA/NMDS analysis. GraphPad Prism 8 was used to plot graphs

**Mouse peritonitis-septicemia model**. To construct an immunosuppressive mice model, the 6–8 weeks old ICR female mice (20 g, $n = 6$ per group, 18 mice were used in total) were treated with 200 mg/kg cyclophosphamide intraperitoneally at the 1$^{st}$ day and 3$^{rd}$ day. After the second cyclophosphamide was treated for 24 h, mice were infected with 0.5 mL of VRE$_{fm}$ CAU427 suspension ($1 \times 10^{10}$ CFU per mouse) via intraperitoneal injection. At 1 h post-infection, all mice were treated with lefamulin and linezolid with 10 mg/kg every 24 h. PBS was used as the negative control. The survival rates of treated mice were recorded during a 96 h period and the bacterial number in the mice organs was counted on the plates.

**Reporting summary**. Further information on research design is available in the Nature Research Reporting Summary linked to this article.

## Data availability
DNA and RNA sequencing data are available in NCBI SRA with accession number PRJNA628015. The MS raw files and proteome sequences data used in this study are available in the Proteome Xchange Consortium under accession code PXD030906. The metabolomic data have been deposited in the CNCB-NGDC repository under accession code PRJCA008528 and in MetaboLights under submission code MTBLS4498. Other data generated in this study are provided in the Supplementary Information and the Source Data file. Source data are provided with this paper.

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

## Acknowledgements

This work is dedicated to Prof. S.Y. Ding for her retirement. We thank all patients for clinical VRE$_{fm}$ isolates in China. This study is supported by the Guangdong Major Project of Basic and Applied Basic Research (2020B0301030007), National Natural Science Foundation of China (31922083), Beijing Municipal Science and Technology (Z201100008920001), and Chinese Universities Scientific Fund (2021RC005).

## Author contributions

J.Z.S. and K.Z. conceived and supervised the project. Q.L. and S.C. performed all the experiments. Q.L. and Y.C.H. conducted the antibacterial tests and animal infection model. S.C. and Q.L. conducted outer-membrane permeability, membrane depolarization, ATP and ROS assays, and membrane fluidity assay. Q.L. performed the transcriptome assay. Z.Q.S., S.Y.D., R.Z., and K.Z. did the data analysis. R.Z., D.X.G., Q.W.Y., H.L.S., F.P.H., H.W., J.C.C. and B.M. shared the clinical VRE$_{fm}$ isolates in this study. Q.L., S.C. and K.Z. wrote the manuscript. All authors read and approved the manuscript.

## Competing interests

The authors declare no competing interests.
