## [Peer Review File · Nature Communications]

Reviewers' Comments:

Reviewer #1:

Remarks to the Author:

The article by Li et al describes collateral sensitivity to pleuromutilins in VREfm. Major findings of the study are that vanA VREfm are sensitive to pleuromutilin antibiotics as opposed to vancomycin-sensitive *E. faecium* which are resistant. The authors of the study demonstrated using an in vivo VREfm colonisation model that the administration of lefamutalin reduced the gastrointestinal burden of VREfm compared to untreated controls and to linezolid treated animals. The authors also found that *msrC* gene expression was significantly reduced in strains carrying the van operon. Using in silico simulations the authors propose that MsrC is able to block pleuromutilin binding to the ribosome leading to the major hypothesis that reduced *msrC* expression in VREfm isolates leads to pleuromutilin sensitivity since the protein can no longer effectively protect the ribosome from the antibiotic. The authors propose that the reduced *msrC* expression observed is caused by binding of vanR to the *msrC* promoter, which they computationally showed has a similar binding sequence to that found in the promoters of the van operon genes.

The findings of this study are interesting and are potentially of clinical relevance. VREfm is an increasingly difficult organism to treat so the discovery that pleuromutilins might be useful is worthy of further evaluation. I believe some of the conclusions in this study are however premature and further experimental validation is required.

1) This study is focused on vanA containing VREfm. While this is the predominant VREfm type in countries such as the USA, vanB-VREfm are also a major clinical problem and in a number of countries such as Australia and New Zealand vanB-VREfm represents the major VREfm type. Differences in the regulation of the vanA and vanB operons in response to external stimuli such as chlorhexidine and daptomycin have also been reported. It is therefore important that the authors evaluate pleuromutilin susceptibility in a range of different vanB-VREfm strains. The clinical relevance of the study would be greatly improved if the authors showed that all VREfm are sensitive to pleuromutilins rather than just vanA-VREfm.

2) The finding that *msrC* expression is higher in VSEfm isolates compared to VREfm coupled with in silico simulations provides compelling data suggesting that MsrC is responsible for pleuromutilin resistance in VSEfm. However, it is not conclusive. To my knowledge previous reports have not linked *msrC* with pleuromutilin resistance. It is most often associated with the MKS phenotype (macrolides, ketolides and group B streptogramins). To conclusively demonstrate that *msrC* is linked with resistance to pleuromutilins it is essential that the authors construct an isogenic *msrC* mutant in a pleuromutilin resistant *E. faecium* genetic background. If the authors' hypothesis is correct then it would be expected that the mutant would revert to sensitivity in comparison to the parental strain. Without this analysis it is not possible to conclusively link *msrC* with pleuromutilin resistance. Whilst I appreciate that the genetic manipulation of *E. faecium* can be difficult, it is far from impossible. There are well documented protocols and molecular tools that can be used to generate isogenic mutants in *E. faecium*.

3) The hypothesis that vanR modulates the expression of *msrC* lacks strong experimental support. As above, isogenic mutants are required to validate this hypothesis. The authors should construct an isogenic vanR mutant in a pleuromutilin sensitive genetic background. If the hypothesis is correct then the isogenic mutant would be expected to revert to resistance in comparison to the parental strain. Transcriptional analysis of the mutant versus the parental strain would also demonstrate that vanR modulates the expression of *msrC*.

4) The suggestion that vanR modulates expression of *msrC* by binding to an operator region within the *msrC* promoter region also requires further validation. I would suggest that an EMSA is needed, whereby the ability of purified vanR to bind to the *msrC* promoter region in question is analysed. This would clearly demonstrate that VanR is able to bind to the *msrC* promoter region as suggested.

5) Testing of lefamutalin in the in vivo mouse model is a nice part of this study. However, for clinical relevance I think the authors should consider testing lefamutalin in a systemic VREfm model

(such as a bacteremia model). This would significantly increase the impact of the findings. The clinical use of pleuromutilins in patients colonised with VREfm seems unlikely. In my opinion, pleuromutilins would be much more likely to be deployed in patients presenting with invasive VREfm infections. Additional data in an invasive VREfm model would therefore provide support for the use of pleuromutilins in these patients.

Overall, I felt this was an interesting study, the manuscript was well written and the data was well presented. However, further experimental validation in several important areas of the study is needed. The lack of isogenic mutants in particular is a major short-coming of the study in its current form.

Reviewer #2:

Remarks to the Author:

The study by Li et al. documents the observation of collateral susceptibility of vancomycin-resistant *E. faecium* (VREfm) to pleuromutilin antibiotics such as lefamulin. The authors used transcriptional profiling to show that this susceptibility tradeoff is due to altered expression of the ribosome protection protein MsrC, which is found to be down-regulated in lefamulin-sensitive VREfm compared to vancomycin-sensitive, lefamulin-resistant *E. faecium* strains. The authors also show that lefamulin can reduce VREfm colonization in a mouse model of intestinal colonization. Overall the study makes several important observations, however a clear explanation for how MsrC becomes down-regulated in VREfm is not provided. The manuscript would be improved by addressing the following comments:

Major comments:

1. It is unclear how the similarity in promoter sequences shown in Fig. 4c causes Phospho-VanR to turn "on" the van operon but turn "off" MsrC. An explanation for this (with supporting experimental evidence) would complete this story, and would greatly increase the impact of this study. The authors state in lines 289-290 that Phospho-VanR induces transcription of the van gene cluster and activates *msrC* transcription, which is inconsistent with their working model of *msrC* down-regulation causing lefamulin susceptibility. This needs to be clarified.

2. Can the authors show a dose-dependent relationship between *msrC* gene or protein abundance and lefamulin susceptibility? This would more clearly implicate *msrC* as the gene responsible for this effect. Alternately, the authors could show that increasing *msrC* expression in a lefamulin-susceptible strain increases lefamulin resistance.

3. The authors state that the collateral response to pleuromutilins is "universal" in VREfm, while Fig. 2d shows that 10% of VREfm isolates are resistant to both vancomycin and pleuromutilins. Proof of increased *msrC* expression in one of these discrepant isolates is provided in extended data Fig. 13, but what about the other 11 isolates tested? Are there genetic differences between these isolates (such as mutations in the *msrC* promoter region) that could be at play in these isolates? The claim of a "universal" effect should also be softened.

Minor comments:

1. The collateral sensitivity scheme presented in Fig. 1a does not convey the intended message, and should be revised. I would suggest removing the left portion of the figure, and breaking out the dose-response curves to show the two steps (resistance to Drug A and collateral sensitivity to Drug B) on separate graphs.

2. Rather than showing MIC₅₀ values in Fig. 2a, the range of MIC values collected for the 40 isolates tested should be shown instead, for example as scatter or box and whisker plots.

3. An explanation for why the resistance ratios differ between the compounds shown in Fig. 2b is needed.

4. Fig. 2d would be more clearly presented as a 2x2 contingency table.

5. Consider swapping Fig. 3 and Fig. 4, to first show the putative mechanism of collateral susceptibility and then the potential utility of lefamulin for decolonization of VREfm.

6. Fig. 4b is not needed and can easily be removed.

7. Error bars are missing from all bar graphs, including those that should have them (Fig. 2c and 4d, for example).

Point-to-Point Response

Reviewer #1 (Remarks to the Author):

Comment:

The article by Li et al describes collateral sensitivity to pleuromutilins in VREfm. Major findings of the study are that vanA VREfm are sensitive to pleuromutilin antibiotics as opposed to vancomycin-sensitive E. faecium which are resistant. The authors of the study demonstrated using an in vivo VREfm colonisation model that the administration of lefamutalin reduced the gastrointestinal burden of VREfm compared to untreated controls and to linezolid treated animals. The authors also found that msrC gene expression was significantly reduced in strains carrying the van operon. Using in silico simulations the authors propose that MsrC is able to block pleuromutilin binding to the ribosome leading to the major hypothesis that reduced msrC expression in VREfm isolates leads to pleuromutilin sensitivity since the protein can no longer effectively protect the ribosome from the antibiotic. The authors propose that the reduced msrC expression observed is caused by binding of vanR to the msrC promoter, which they computationally showed has a similar binding sequence to that found in the promoters of the van operon genes.

The findings of this study are interesting and are potentially of clinical relevance. VREfm is an increasingly difficult organism to treat so the discovery that pleuromutilins might be useful is worthy of further evaluation. I believe some of the conclusions in this study are however premature and further experimental validation is required.

Response: We thank the reviewer for recognizing the novelty and significance of the work. We are grateful for the insightful comments/suggestions. In light of these comments, we have provided additional experimental results and thoroughly revised the manuscript to clarify methods, data analysis, and interpretation to improve rigor.

1) This study is focused on vanA containing VREfm. While this is the predominant VREfm type in countries such as the USA, vanB-VREfm are also a major clinical

problem and in a number of countries such as Australia and New Zealand *vanB*-VREfm represents the major VREfm type. Differences in the regulation of the *vanA* and *vanB* operons in response to external stimuli such as chlorhexidine and daptomycin have also been reported. It is therefore important that the authors evaluate pleuromutilin susceptibility in a range of different *vanB*-VREfm strains. The clinical relevance of the study would be greatly improved if the authors showed that all VREfm are sensitive to pleuromutilins rather than just *vanA*-VREfm.

Response: Thank you for the insightful comments and suggestions. We agree that there are differences in the regulation of the *vanA* and *vanB* operons. Given that either *vanS_A* or *vanS_B* induces P-VanR binding to the promoter of *msrC* (*Antimicrob Agents Chemother*, **2016**, 60, 2209), we first compared the promoter sequences of *vanR* and *msrC* in *vanB*-type isolates worldwide based on whole-genome sequence analysis. These isolates share the same promoters (**Fig. R1**), suggesting that *vanB*-type *E. faecium* may show collateral sensitivity to pleuromutilins as well.

Fig. R1 The *vanA*- and *vanB*- type *E. faecium* share the same patterns in the promoters of *msrC*/*vanR*. CAU369, VREF001, ISMMVRE-1, ERV196, and AUS04005 are *vanA*-type isolates; CAU996, *E. faecium* 8672, MLG856-2, AE12, Efm 6123 are *vanB*-type isolates.

Subsequently, we determined the susceptibility of five pleuromutilins against a clinically isolated *vanB*-VRE_{fm} strain. Consistent with that in the majority of *vanA*-VRE_{fm} strains, *vanB* VRE_{fm} CAU996 shows collateral sensitivity to all pleuromutilins tested (**Table R1**). Furthermore, we observed that the increased transcription of either *vanR* or *vanS* in a dose-dependent manner, whereas the decreased *msrC* in both *vanA* and *vanB* VRE_{fm} isolates (**Fig. R2**). These results indicate that low expression of *msrC* potentiates the activity of pleuromutilins against *vanB* VRE_{fm}, consistent with the collateral response to pleuromutilins in *vanA* VRE_{fm} isolates. Altogether, these results indicate that *vanB* VRE_{fm} exhibits similar collateral sensitivity to pleuromutilins as *vanA* VRE_{fm}.

Table R1 MICs of multiple antibiotics in *vanA*- and *vanB*- type *E. faecium* (μg/mL).

	Pleuromutilin antibiotics						
	LMU	RET	VAL	TIA	AZA	VAN	LZD
E. faecium CAU996 (vanB)	<0.03	<0.03	<0.03	0.125	0.125	>128	2
E. faecium CAU369 (vanA)	<0.03	<0.03	<0.03	0.125	0.125	>128	1

LMU: Lefamulin; RET: Retapamulin; VAL: Valnemulin; TIA: Tiamulin; AZA: Azamulin; VAN: Vancomycin; LZD: Linezolid.

Fig. R2 Transcription of *vanR/vanS/msrC* in VRE_{fm} in the presence of lefamulin. Transcription analysis of *msrC/vanR/vanS* in *vanA*-type VRE_{fm} CAU369 (**a, b, c**) and *vanB*-type VRE_{fm} CAU996 (**d, e, f**). Both VRE_{fm} isolates were treated with lefamulin at different concentrations for 1 h. Experiments were performed as three biologically independent experiments, and the mean \pm S.D. (n=3) were shown. *P* values were determined by non-parametric one-way ANOVA.

However, we apologize for that we can not include more *vanB*-type *E. faecium* here. Because the main epidemic strain in China is not *vanB*, it is very hard to collect more (*Front Med*, **2020**, 8, 403; *J Microbiol Immunol Infect*, **2020**, 53, 746). We obtained only one *vanB*-type VRE_{fm} strain from more than 100 VRE_{fm} isolates, others are *vanA*- and/or *vanM*-type. Meanwhile, it is difficult to get more *vanB*-type VRE_{fm} from other countries due to the COVID-19 pandemic. We will be happy to test the efficacy and generality of pleuromutilins against *vanB*-type VRE_{fm} in the future.

Fig. R1, **Fig. R2c**, and **Table R1** have been updated as **Extended Data Fig. 15**, **Fig. 3a** and **Fig. 1b** in the revised manuscript, respectively.

2) *The finding that msrC expression is higher in VSEfm isolates compared to VREfm coupled with in silico simulations provides compelling data suggesting that MsrC is responsible for pleuromutilin resistance in VSEfm. However, it is not conclusive. To my knowledge previous reports have not linked msrC with pleuromutilin resistance. It is most often associated with the MKS phenotype (macrolides, ketolides and group B streptogramins). To conclusively demonstrate that msrC is linked with resistance to pleuromutilins it is essential that the authors construct an isogenic msrC mutant in a pleuromutilin resistant E. faecium genetic background. If the authors' hypothesis is correct then it would be expected that the mutant would revert to sensitivity in comparison to the parental strain. Without this analysis it is not possible to conclusively link msrC with pleuromutilin resistance. Whilst I appreciate that the genetic manipulation of E. faecium can be difficult, it is far from impossible. There are well documented protocols and molecular tools that can be used to generate isogenic mutants in E. faecium.*

Response: Thank you for the insightful comments and great suggestions. Previous studies show that MsrC, belonging to the ATP-binding cassette F (ABC-F) protein family, binds to the peptidyl transferase center (PTC) of ribosome (*mBio*, **2016**, 7, e01975). ABC-F proteins protect the PTC, resulting in cross-resistance to antibiotics that target PTC (*PNAS*, **2018**, 115, 5157; *Nat Rev Microbiol*, **2020**, 18, 637). For example, MsrC binds to the PTC (**Fig. 3a**), which are similar to the binding sites of pleuromutilins to PTC at the sites of U2504 and C2063 regions (*Nat Rev Microbiol*, **2014**, 12, 35). Therefore, we hypothesized that MsrC is associated with pleuromutilins resistance. Toward this end, we first supplied additional assay based on proteomics analysis. Remarkably, we found that the increased expression of MsrC in VRE_{fm} treated with lefamulin for 1 h (**Fig. R3**), indicating that MsrC may be responsible for pleuromutilin resistance.

Fig. R3 High expression of MsrC induces the resistance to pleuromutilins in VRE_{fm}. (a) Volcano plot represents protein expression ratios in lefamulin treated VRE_{fm} CAU378. For each protein, the $-\log_{10}$ (P-value) is plotted against its \log_2 (fold change). Upregulated proteins ($P < 0.05$, fold change > 2) in the presence of 1× and 10×MIC lefamulin are coloured in red, while downregulated proteins ($P < 0.05$, fold change < -2) are coloured in blue. (b) Mean value of \log_2 [fold change] of relative expression of MsrC. Proteomics analysis of VRE_{fm} under the treatment of lefamulin at the levels of 1× and 10×MIC for 1 h.

Furthermore, we constructed a *msrC* overexpression mutant in a pleuromutilins sensitive strain based on a modified conjugative transformation assay (*J Appl Microbiol*, **2020**, 129, 565). First, we transformed a recombinant plasmid pAM401+*msrC* from the donor *E. faecalis* JH2-2 into *E. faecium* GE-1, and obtained the mutant selected by chloramphenicol and arabinose (**Fig. R4a, b**). The mutant of interest can be distinguished by arabinose from the donor due to the metabolic preference (*J Clin Microbiol*, **1994**, 32, 2999; *Mol Biotechnol*, **2019**, 61, 385), despite that they share similar drug-resistance patterns. The increased expression of *msrC* in mutants treated with lefamulin is in a dose-dependent manner, denoting that we successfully constructed a *msrC* overexpressed strain (**Fig. R4c**).

Fig. R4 The overexpression of *msrC* induces pleuromutilins resistance.

- (a-b) Design of *msrC* expression plasmid (a) and scheme of transformation (b).
(c) The expression of *msrC* in the mutant (pAM401+*msrC*) in the presence of a concentration gradient of lefamulin.
(d) Comparison of antibiotic susceptibility in wild-type *E. faecium* and mutant.

Then, we determined the susceptibility of five pleuromutilins against the mutant. The conjugant shows resistance to all pleuromutilins and antibiotics targeting ribosome (**Fig. R4d**). Notably, the mutant is dramatically resistant to lincomycin and erythromycin (**Table R2**), consistent with previous observations that ABC-F proteins particularly Msr-type proteins prevent the PTC of ribosome from antibiotic stresses (*PNAS*, **2018**, 115, 5157; *Annu Rev Biochem*, **2018**, 87, 451; *Nat Rev Microbiol*, **2020**, 18, 637). Altogether, these results indicate that high expression of *msrC* is linked with the resistance to pleuromutilins.

Table R2 MICs of multiple antibiotics in wild-type *E. faecium* and mutant ($\mu\text{g/mL}$).

	Pleuromutilin antibiotics					VAN	LIN	ERY
	LMU	RET	VAL	TIA	AZA			
Wild-type	0.06	0.06	0.06	0.25	0.25	2	1	1
Mutant (pAM401+ msrC)	16	16	16	32	32	2	>128	>128

LMU: Lefamulin; RET: Retapamulin; VAL: Valnemulin; TIA: Tiamulin; AZA: Azamulin; VAN: Vancomycin; LIN: Lincomycin; ERY: Erythromycin.

In addition, we tried to design a suicide plasmid to knockout *msrC* in pleuromutilin resistant *E. faecium*s isolates (**Fig. R5a, b**). Although the frequency of conjugation is very low, we recently have obtained 10 mutants carrying suicide plasmids. The recombinant suicide plasmid was successfully constructed and removed in *E. coli* after overnight culture at 42 °C (**Fig. R5c**). And we found the expression of *cat* in *E. faecium* carrying such recombinant suicide plasmids (**Fig. R5d**), however, the $\Delta\textit{msrC}$ mutant could not be collected under this condition. We deduce that the maintenance of thermosensitive-plasmids in *E. faecium* at 42 °C maybe caused by the unique metabolic pattern of *E. faecium*, such as the lack of the tricarboxylic acid cycle (*Nat Microbiol*, **2019**, 4, 1716; *BMC Genomics*, **2010**, 11, 239). Currently, we are continuing to optimize the experimental conditions and exploit more suitable suicide plasmids, to knockout the *msrC* gene in *E. faecium*.

Fig. R5 Vector construction and scheme of transfection.

(a-b) Vector construction. Design of suicide plasmid (a) and scheme of transformation (b).

(c) Screening the conjugant removing the suicide plasmid in *E. coli* after overnight culture at 42 °C. The conjugant loses chloramphenicol resistance after suicide plasmid removing.

(d) Screening the conjugant carrying suicide plasmid in *E. faecium* at after overnight culture 42 °C, 45 °C and 48 °C. The conjugant shows chloramphenicol resistance, but fails to remove suicide plasmid.

Fig. R3, R4, and Table R2 have been updated as Fig 4c-d, Extended Data Fig. 12 and Extended Data Table 8 in the revised manuscript, respectively.

3) The hypothesis that *vanR* modulates the expression of *msrC* lacks strong experimental support. As above, isogenic mutants are required to validate this hypothesis. The authors should construct an isogenic *vanR* mutant in a pleuromutilin sensitive genetic background. If the hypothesis is correct then the isogenic mutant would be expected to revert to resistance in comparison to the parental strain. Transcriptional analysis of the mutant versus the parental strain would also demonstrate that *vanR* modulates the expression of *msrC*.

Response: Thank you for the insightful comments and great suggestions. To validate that *vanR* modulates the expression of *msrC*, we first constructed a conjugant by receiving a recombinant *vanRS* plasmid in a pleuromutilin resistant *E. faecium* (**Fig. R6**). The transcription of *vanR* and *vanS* were activated in a dose-dependent manner under lefamulin treatment (**Fig. R7a, b**), in turn, the transcription of *msrC* in the conjugant (pAM401+ *vanRS*) was inhibited (**Fig. R7c**). Correspondingly, the conjugant with *vanRS* expression is sensitive to pleuromutilins, with more than 16-fold decreased MICs (**Fig. R7d, Table R3**). Meanwhile, we observed the delayed growth curves of conjugant-2 with slight fitness cost in the presence of a subinhibitory level of lefamulin, compared to that of conjugant-1 (**Fig. R8**). Collectively, these results support our claim that *vanR* modulates the expression of *msrC* in VRE_{fm}.

Fig. R6 Vector construction and transfection scheme.

(a) Vector construction. Design of *vanRS* mutant plasmid.

(b) Transfection scheme of electroporation and conjugative transformation.

Conjugant-2 (pAM401+ *vanRS*) was screened through this pathway.

(c) Screening the conjugant carrying pAM401+ *vanRS* plasmid in *E. faecium*. The conjugant shows chloramphenicol resistance.

(d) PCR amplification products of the conjugant. VRE_{fm} CAU369 and pAM401 were used as controls.

Fig. R7 The *vanRS* inhibits the transcription of *msrC*.
(a-b) Transcription of *vanR* **(a)** and *vanS* **(b)** in the conjugant treated with lefamulin.
(c) Comparison of *msrC* transcription in wild type *E. faecium* and conjugant in the presence of lefamulin.
(d) Comparison of the MICs of multiple antibiotics in wild type *E. faecium* and conjugant. Vancomycin and linezolid were used as controls.

Fig. R8 Lefamulin inhibited the growth of the conjugant carrying *vanRS*.
 Growth curves of wild type *E. faecium* and conjugants treated with a sublethal level of lefamulin (1/2 MIC). Conjugant-1 receives sole plasmid pAM401, whereas Conjugant-2 receives the plasmid pAM401+*vanRS*.

Experiments were performed as three biologically independent experiments, and the mean \pm S.D. (n=3) were shown. *P* values were determined by non-parametric one-way ANOVA.

Fig. R6, R7, R8 have been updated as **Extended Data Fig. 18, 19, 20** in the revised manuscript, respectively.

4) *The suggestion that vanR modulates expression of msrC by binding to an operator region within the msrC promoter region also requires further validation. I would suggest that an EMSA is needed, whereby the ability of purified vanR to bind to the msrC promoter region in question is analysed. This would clearly demonstrate that VanR is able to bind to the msrC promoter region as suggested.*

Response: Thank you for the great suggestion. To validate that P-VanR regulates *msrC* transcription, we first purified *vanR* and prepared P-VanR, according to a previously described protocol (*Biochemistry*, **1994**, 33, 4625). Subsequently, we demonstrated that P-VanR binds to *msrC* promoter by electrophoretic mobility shift assay (**Fig. R9a, b**). We calculate the EC₅₀ (effective concentration for 50% response) as 1.01 μ mol/L for P-VanR binding to the *msrC* promoter fragment (**Fig. R9c**), which is much lower than the binding affinity between P-VanR and the *vanH* promoter. Altogether, these results indicate that P-VanR co-regulates *vanR/vanH* and *msrC* by binding to similar promoter fragments, facilitating pleuromutilins against VRE_{fm}.

Fig. R9 P-VanR binds to *msrC* promoter fragment.

(a) Binding reactions between P-VanR (12-1.5 μ M) and *msrC* promoter fragment (212-bp, 0.3 ng) based on the gel electrophoretic mobility shift assay. Free probe: biotin-labeled promoter; Competitor: Unlabeled promoter; BSA: Bovine serum albumin.

(b) Calculated protein-binding rates of P-VanR and *vanH/vanR/msrC* promoters,

based on the gray values in **Fig. R9a**.

(c) Binding constants of *vanH/vanR/msrC* promoters and P-VanR.

Fig. R9 has been updated as **Fig. 4e-g** in the revised manuscript.

5) Testing of lefamutilin in the in vivo mouse model is a nice part of this study.

However, for clinical relevance I think the authors should consider testing lefamutilin in a systemic VRE_{fm} model (such as a bacteremia model). This would significantly increase the impact of the findings. The clinical use of pleuromutilins in patients colonised with VRE_{fm} seems unlikely. In my opinion, pleuromutilins would be much more likely to be deployed in patients presenting with invasive VRE_{fm} infections. Additional data in an invasive VRE_{fm} model would therefore provide support for the use of pleuromutilins in these patients.

Response: Thank you for the suggestions. We agree that the efficacy of lefamulin in a systemic VRE_{fm} infection model will be more promising. Given the fact that most VRE_{fm} are hypovirulent (*Nat Rev Microbiol*, **2012**, 10, 266; *Infect Drug Resist*, **2015**, 8, 217.), we first screened the virulence genes in VRE_{fm} clinical isolates (n = 109) based on informatics analysis. Three isolates carry a series of virulence genes (**Fig. R10a**), consistent with previous reports (*Nat Rev Microbiol*, **2012**, 10, 266; *Foods*, **2021**, 10, 2846). Interestingly, we found that these diverse virulence genes are not the prerequisite for the lethality in mice. Compared to the other isolates tested in an immunosuppressive peritonitis-septicemia model, remarkably, VRE_{fm} CAU427 (1×10^{10} CFUs/mouse) effectively led to the death of all mice in four days (**Fig. R10b**). Therefore, we used VRE_{fm} CAU427 as a model strain to evaluate the efficacy of lefamulin in the peritonitis-septicemia model.

Fig. R10 Screening of VRE_{fm} for a systemic infection.

(a) The virulence genes in VRE_{fm} clinical isolates.

(b) The survival rate of VRE_{fm} infected mice in an immunosuppressive peritonitis-septicemia model (n = 6). Each VRE_{fm} was intraperitoneally injected into mice with 1×10^{10} CFUs.

Both lefamulin and linezolid were intraperitoneally injected every 24 h (**Fig. R11a**) to treat infected mice. The survival curve showed all mice survived under the treatment of lefamulin with a dose of 10 mg/kg after 96 h (**Fig. R11b**), and the bacterial counts in different organs dramatically reduced (**Fig. R11c**). All these results demonstrate that lefamulin effectively treat a systemic VRE_{fm} infection.

Fig. R11 The efficacy of lefamulin in a VRE_{fm} peritonitis-septicemia model.

(a) Scheme of VRE_{fm} peritonitis-septicemia model.

(b) Survival rates of peritonitis-septicemia mice infected with VRE_{fm} CAU427 (1.0×10^{10} CFUs) in the presence of PBS, lefamulin and linezolid.

(c) Lefamulin (10 mg/kg) decreased bacterial loads in different organs of mice in the

mouse peritonitis-sepsis model.

Fig. R11 has been updated as **Fig. 5d-f** in the revised manuscript.

Reviewer #2 (Remarks to the Author):

Comment:

*The study by Li et al. documents the observation of collateral susceptibility of vancomycin-resistant *E. faecium* (VREfm) to pleuromutilin antibiotics such as lefamulin. The authors used transcriptional profiling to show that this susceptibility tradeoff is due to altered expression of the ribosome protection protein MsrC, which is found to be down-regulated in lefamulin-sensitive VREfm compared to vancomycin-sensitive, lefamulin-resistant *E. faecium* strains. The authors also show that lefamulin can reduce VREfm colonization in a mouse model of intestinal colonization. Overall the study makes several important observations, however a clear explanation for how MsrC becomes down-regulated in VREfm is not provided.*

Response: We thank the reviewer for recognizing the novelty and significance of the work. We are grateful for the reviewer's insightful comments/suggestions to improve the manuscript. In light of these comments, we have provided additional experimental information and revised the manuscript to clarify methods, data analysis, and interpretation to improve rigor.

Major comments:

1. It is unclear how the similarity in promoter sequences shown in Fig. 4c causes Phospho-VanR to turn "on" the van operon but turn "off" MsrC. An explanation for this (with supporting experimental evidence) would complete this story, and would greatly increase the impact of this study. The authors state in lines 289-290 that Phospho-VanR induces transcription of the van gene cluster and activates msrC transcription, which is inconsistent with their working model of msrC down-regulation causing lefamulin susceptibility. This needs to be clarified.

Response: Thank you for the insightful comments and great suggestions. To validate that P-VanR regulates *msrC* transcription, we first purified VanR and prepared P-VanR, according to a previously described protocol (*Biochemistry*, **1994**, 33, 4625). Subsequently, we demonstrate that P-VanR binds to *msrC* promoter by a gel

electrophoretic mobility shift assay (**Fig. R1a, b**). Meanwhile, we calculate the EC₅₀ (effective concentration for 50% response) as 1.01 μmol/L for P-VanR binding to the *msrC* promoter fragment (**Fig. R1c**), which is much lower than the binding affinity between P-VanR and the *vanH* promoter. Altogether, these results indicate that P-VanR co-regulates *vanR/vanH* and *msrC* by binding to similar promoter fragments, facilitating pleuromutilins against VRE_{fm}.

We apologize for the confusion in lines 289-290 and have revised the state as “*Phospho-VanR induces transcription of the van gene cluster and inhibits msrC transcription*”, in the revised line 276, page 11.

Fig. R1 P-VanR binds to *msrC* promoter fragment.

(a) Binding reactions between P-VanR (12-1.5 μM) and *msrC* promoter fragment (212-bp, 0.3 ng) based on the gel electrophoretic mobility shift assay. Free probe: biotin-labeled promoter; Competitor: Unlabeled promoter; BSA: Bovine serum albumin.

(b) Calculated protein-binding rates of P-VanR and *vanH/vanR/msrC* promoters, based on the gray values in **Fig. R1a**.

(c) Binding constants of *vanH/vanR/msrC* promoters and P-VanR.

Fig. R1 has been updated as **Fig. 4e-g** in the revised manuscript accordingly.

2. Can the authors show a dose-dependent relationship between *msrC* gene or protein abundance and lefamulin susceptibility? This would more clearly implicate *msrC* as the gene responsible for this effect. Alternately, the authors could show that increasing *msrC* expression in a lefamulin-susceptible strain increases lefamulin resistance.

Response: Thank you for the suggestions. We agree that the quantitative relationship between MsrC and lefamulin susceptibility is required. We first observed the increased expression of MsrC in a lefamulin resistant VRE_{fm} CAU378 treated with lefamulin for 1 h, based on proteomics analysis (**Fig. R2**). Constantly, we found that the increased transcription of *msrC* in two lefamulin resistant isolates (**Fig. R3a-b**) is in a dose-dependent manner of lefamulin. Meanwhile, we observed decreased *msrC* transcription of *msrC* in two lefamulin sensitive isolates as well (**Fig. R3c-d**).

Fig. R2 High expression of MsrC induces the resistance to pleuromutilins in VRE_{fm}. (a) Volcano plot representing protein expression ratios of the lefamulin treated bacterial cells. For each protein, the -log₁₀ (P-value) is plotted against its log₂ (fold change). Proteins upregulated (P < 0.05, fold change >2) in 1× and 10×MIC lefamulin treated samples are coloured in red, proteins downregulated (P < 0.05, fold change < -2) are coloured in blue while unchanged in black. (b) Proteomics analysis of VRE_{fm} CAU378 treated with 1× and 10×MIC lefamulin for 1h. Proteins were identified as significantly different with fold changes of log₂ [fold changes] values of at fold increase or fold decrease at expression levels. (c) Mean value of log₂ [fold change] of relative expression of MsrC and VanS. Proteomics analysis of VRE_{fm} CAU378 under the treatment of lefamulin at the levels of 1× and 10×MIC for 1 h.

Pleuromutilin resistant VRE_{fm}

Pleuromutilin sensitive VRE_{fm}

Fig. R3 Transcription of *msrC* in VRE_{fm} in the presence of lefamulin. Transcription analysis of *msrC* in pleuromutilin resistant isolates (**a-b**) and sensitive isolates (**c-d**). Both VRE_{fm} isolates were treated with lefamulin for 1 h. Experiments were performed as three biologically independent experiments, and the mean \pm S.D. (n=3) were shown. *P* values were determined by non-parametric one-way ANOVA.

Furthermore, we constructed a MsrC overexpression mutant in a pleuromutilin sensitive strain based on a modified conjugative transformation assay (*J Appl Microbiol.*, **2020**, 129, 565). First, we transformed a recombinant plasmid pAM401+*msrC* from the donor *E. faecalis* JH2-2 into *E. faecium* GE-1, and obtained the mutant selected by chloramphenicol and arabinose (**Fig. R4a, b**). The mutant of interest can be distinguished by arabinose from the donor due to the metabolic preference (*J Clin Microbiol*, **1994**, 32, 2999; *Mol Biotechnol*, **2019**, 61, 385), despite that they share similar drug-resistance patterns. The increased expression of *msrC* in mutants treated with lefamulin is in a dose-dependent manner, denoting that we

successfully constructed a *msrC* overexpressed strain (**Fig. R4c**). Then, we determined the susceptibility of five pleuromutilins against the mutant. The conjugant shows resistance to all pleuromutilins and antibiotics targeting ribosome (**Fig. R4d**). Notably, the mutant is dramatically resistant to lincomycin and erythromycin, consistent with previous observations that ABC-F proteins particularly Msr-type proteins prevent the PTC of ribosome from antibiotic stresses (*PNAS*, **2018**, 115, 5157; *Annu Rev Biochem*, **2018**, 87, 451; *Nat Rev Microbiol*, **2020**, 18, 637). Altogether, these results indicate that there is a dose-dependent relationship between the transcription of *msrC* and lefamulin susceptibility.

Fig. R4 The overexpression of *msrC* induces pleuromutilins resistance. (a, b) Design of *msrC* expression plasmid (a) and scheme of transformation (b). (c) The expression of *msrC* in the mutant (pAM401+ *msrC*) in the presence of a concentration gradient of lefamulin. (d) Comparison of antibiotic susceptibility in wild-type *E. faecium* and mutant.

Fig. R2, R3 and R4 have been updated as **Fig. 3c, Fig. 3a-b** and **Extended Data Fig. 12** in the revised manuscript, respectively.

3. The authors state that the collateral response to pleuromutilins is "universal" in VRE_{fm}, while Fig. 2d shows that 10% of VRE_{fm} isolates are resistant to both vancomycin and pleuromutilins. Proof of increased *msrC* expression in one of these discrepant isolates is provided in extended data Fig. 13, but what about the other 11 isolates tested? Are there genetic differences between these isolates (such as mutations in the *msrC* promoter region) that could be at play in these isolates? The claim of a "universal" effect should also be softened.

Response: Thank you for the insightful comments and suggestions. We are sorry for the misunderstanding and have revised “universal” to “most”.

To dissect the phenomenological observations, we first confirmed that there were no mutations at such sites in these isolates based on whole-genome sequence analysis (**Fig. R5**). Consisting with our previous observation of increased *msrC* expression, we found a dose dependent increase of *msrC* expression in all 12 isolates (**Fig. R3a-b**, **Fig. R6**), implying that increased expression of *msrC* is highly responsible for the resistance to pleuromutilins in VRE_{fm}.

Fig. R5 No mutations in *msrC* promoter regions. All 12 VRE_{fm} isolates are resistant to pleuromutilins. The elements of promoter are in red and *msrC* is in gray.

Fig. R6 Transcription of *msrC* in pleuromutilins resistant VRE_{fm} isolates increased in a dose-dependent manner. Transcription analysis of *msrC* in pleuromutilins resistant VRE_{fm} isolate (a-i) VRE_{fm} isolates were treated with lefamulin for 1 h. Experiments were performed as three biologically independent experiments, and the mean \pm S.D. (n=3) is shown. *P* values were determined by non-parametric one-way ANOVA.

The key points of our manuscript are clear. We mainly focus on the collateral response to pleuromutilins in most VRE_{fm}, however, the underlying mechanism of partial VRE_{fm} strains resistant to pleuromutilins remains elusive. Based on the proteomics analysis, VanS shows no change in the pleuromutilin-resistant VRE_{fm} CAU378 (Fig. R2a, c), suggesting that low expression of VanS may reduce the phosphorylation of VanR. Meanwhile, given that P-VanR shows approximate 14-folds higher affinity to the promoter of *vanH* than *msrC* (Fig. R1c), we deduce that the insufficient P-VanR preferentially bind to the *vanR/vanH* promoters, resulting in the abolition of *msrC* inhibition to cause pleuromutilins resistance.

However, we should emphasize that, regardless of the underlying mechanism for

these minority pleuromutilin-resistant VRE_{fm} isolates, it does not affect our central point: VRE_{fm} show collateral sensitivity to pleuromutilins.

Fig. R6 has been updated as **Extended Data Fig. 11** in the revised manuscript.

Minor comments:

1. *The collateral sensitivity scheme presented in Fig. 1a does not convey the intended message, and should be revised. I would suggest removing the left portion of the figure, and breaking out the dose-response curves to show the two steps (resistance to Drug A and collateral sensitivity to Drug B) on separate graphs.*

Response: Thank you for the suggestion. We have revised the scheme of collateral sensitivity (**Fig. R7**). **Fig. R7** has been updated as **Fig. 1a** in the revision.

Fig. R7 Scheme of collateral sensitivity. Bacterial evolution of resistance to one antibiotic is usually accompanied by collateral sensitivity to another antibiotic.

2. *Rather than showing MIC₅₀ values in Fig. 2a, the range of MIC values collected for the 40 isolates tested should be shown instead, for example as scatter or box and whisker plots.*

Response: Thank you for the suggestion. **Fig. 2a** has been updated as **Fig. R8** in the revision

Fig. R8 MIC₅₀ of seven ribosome-targeting antibiotics against *E. faecium* (n = 40).

3. An explanation for why the resistance ratios differ between the compounds shown in Fig. 2b is needed.

Response: Thank you for the suggestion. The resistance ratios are calculated by MIC₅₀ (VRE)/MIC₅₀ (VSE). In the previous manuscript, we obtained the MICs of azamulin and tiamulin with maximum concentration of 16 µg/mL, and the MIC₅₀ (VSE) was great than 16 µg/mL. Then we calculated the RR of azamulin and tiamulin with MIC₅₀ (VSE) of 16 µg/mL, which is unprecise. Therefore, we performed new assays for azamulin and tiamulin again and extended the maximum concentrations from 16 µg/mL to 128 µg/mL. The revised RR of azamulin and tiamulin are 0.00195 (**Fig. R9**), similar to that of retapamulin, valnemulin and lefamulin. There are no significant difference of RR among these compounds. **Fig. 2b** has been updated as **Fig. R9** in the revision

Fig. R9 Resistance ratios of pleuromutilins against 210 *E. faecium* isolates. Structures of five pleuromutilins, where the C19 of the tricyclic mutilin core is single bond in azamulin.

4. Fig. 2d would be more clearly presented as a 2x2 contingency table.

Response: Thank you for the suggestion. We have updated the figure format accordingly, and **Fig. 2d** has been updated as **Fig. R10** in the revision.

Phenotype rate		Pleuromutilins	
		P_s	P_r
Vancomycin	V_s	11.9% (13/109)	88.1% (89/109)
	V_r	88.2% (90/102)	11.8% (12/102)

Fig. R10 Proportion of four phenotypes in *E. faecium*. ~90% VRE_{fm} ($n = 102$) are sensitive to pleuromutilins, whereas ~90% of VSE_{fm} ($n = 109$) are resistant. V_s : sensitivity to vancomycin, V_r : resistance to vancomycin, P_s : sensitivity to pleuromutilins, P_r : resistance to pleuromutilins.

5. Consider swapping Fig. 3 and Fig. 4, to first show the putative mechanism of collateral susceptibility and then the potential utility of lefamulin for decolonization of VRE_{fm} .

Response: Thank you for the insightful comments and suggestions. We have updated the order of figures according to your suggestion.

6. Fig. 4b is not needed and can easily be removed.

Response: Thank you for the suggestion. We have removed the Fig. 4b in the revision.

7. Error bars are missing from all bar graphs, including those that should have them (Fig. 2c and 4d, for example).

Response: We have updated the error bars in all of graphs accordingly.

Reviewers' Comments:

Reviewer #1:

Remarks to the Author:

I have no further concerns relating to this work. The experiments I requested previously have been performed to my satisfaction and experimental questions addressed. It is my belief that the conclusions drawn in this manuscript are now scientifically sound and the experimental findings are much more robust than in the previous iteration of the work.

I applaud the authors for the effort that has clearly been put in to do all of the extra experimentation requested.

Reviewer #2:

Remarks to the Author:

The revised manuscript has addressed my concerns. I have two very minor final comments:

1. Fig. 2c - "faeciums" should not be plural.
2. Fig. 2d - There seems to be a mistake in the Vs/Pr box, I think this should be 96/109, but please check the math for all four entries in the table.

Point-to-Point Response

Reviewer #1 (Remarks to the Author):

Comment:

I have no further concerns relating to this work. The experiments I requested previously have been performed to my satisfaction and experimental questions addressed. It is my belief that the conclusions drawn in this manuscript are now scientifically sound and the experimental findings are much more robust than in the previous iteration of the work.

I applaud the authors for the effort that has clearly been put in to do all of the extra experimentation requested.

Response: We thank the reviewer for recognizing the novelty and significance of the work.

Reviewer #2 (Remarks to the Author):

Comment:

The revised manuscript has addressed my concerns. I have two very minor final comments:

1. Fig. 2c - "faeciums" should not be plural.

Response: Thank you for the suggestion. We have updated Fig. 2c - "faeciums" into "faecium".

2. Fig. 2d - There seems to be a mistake in the Vs/Pr box, I think this should be 96/109, but please check the math for all four entries in the table.

Response: Thank you for the suggestion. We have corrected the mistake and checked the math in the table.